# CLARA-A3: The third edition of the AVHRR-based CM SAF climate data record on clouds, radiation and surface albedo covering the period 1979 to 2023

Karl-Göran Karlsson[1], Martin Stengel[2], Jan Fokke Meirink[3], Aku Riihelä[4], Jörg Trentmann[2], Tom Akkermans[5], Diana Stein[2], Abhay Devasthale[1], Salomon Eliasson[1], Erik Johansson[1], Nina Håkansson[1], Irina Solodovnik[2], Nikos Benas[3], Nicolas Clerbaux[5], Nathalie Selbach[2], Marc Schröder[2] and Rainer Hollmann[2]

[1]Meteorological Research Unit, Research and Development Department, The Swedish Meteorological and Hydrological Institute (SMHI), Folkborgsvägen 17, 602 10 Norrköping, Sweden
[2]Satellite-Based Climate Monitoring, Deutscher Wetterdienst, Frankfurter Str. 135, 63067 Offenbach am Main, Germany
[3]R&D Satellite Observations, Royal Netherlands Meteorological Institute (KNMI), Utrechtseweg 297, 3731 GA De Bilt, The Netherlands
[4]Meteorological Research, Finnish Meteorological Institute, Helsinki, 00101, Finland
[5]Remote Sensing from Space, Royal Meteorological Institute of Belgium, B-1180 Brussels, Belgium

*Correspondence to*: Karl-Göran Karlsson (karl-goran.karlsson@smhi.se)

**Abstract.**

This paper presents the third edition of the CM SAF cLoud, Albedo and surface RAdiation dataset from AVHRR data, CLARA-A3. The content of earlier CLARA editions, namely cloud, surface albedo, and surface radiation products, has been extended with two additional surface albedo products (blue and white sky albedo), three additional surface radiation products (net shortwave and longwave radiation, surface radiation budget) and two top of atmosphere radiation budget products (reflected solar flux and outgoing longwave radiation). The record length is extended to 42 years (1979-2020) by also incorporating results from the first version of the Advanced Very High Resolution Radiometer (AVHRR) imager (AVHRR/1). A continuous extension of the climate data record (CDR) has also been implemented by processing an interim climate data record (ICDR) based on the same set of algorithms but with slightly changed ancillary input data. All products are briefly described together with validation results and inter-comparisons with currently existing similar CDRs. The extension of the product portfolio and the temporal coverage of the data record, together with product improvements, is expected to enlarge the potential of using CLARA-A3 for climate change studies and, in particular, studies of potential feedback effects between clouds, surface albedo and radiation.

The CLARA-A3 data record is hosted by the European Organisation for the Exploitation of Meteorological Satellites (EUMETSAT) Satellite Application Facility on Climate Monitoring (CM SAF) and is freely available at https://doi.org/10.5676/EUM_SAF_CM/CLARA_AVHRR/V003.

**1 Introduction**

Monitoring clouds, radiation, and surface conditions on the Earth is essential for understanding climate and how climate
changes. In particular, changes in cloudiness, cloud properties, and surface albedo are key elements in understanding the main
drivers of climate change, namely changes to the Earth's radiation balance, both at the surface and at the top of atmosphere
(TOA).

Climate monitoring requires global coverage and observations over a long time (i.e., several decades). Only satellite
observations can provide global coverage, but their temporal coverage is still relatively short from the climate perspective.
However, satellite-based CDRs have been compiled for quite some time, with the International Satellite Cloud Climatology
Project (ISCCP) acting as a pioneer (Schiffer and Rossow, 1981 and Rossow and Schiffer, 1990). Additional CDRs have been
compiled as time series of individual satellite sensors have grown in length, and new sensors have become available. Two
examples are the Pathfinder Atmospheres – Extended (PATMOS-x, Heidinger et al., 2014) CDR for cloud and radiation
properties and the Moderate Resolution Imaging Spectroradiometer (MODIS) CDRs for cloud, radiation and surface properties
(e.g., Platnick et al., 2015). In addition, measurements from the Visible Infrared Imaging Radiometer Suite (VIIRS) instrument
is now capable of providing decadal scale observations suitable for CDR generation and they will also be used to extend the
MODIS CDR through a specific VIIRS+MODIS continuity product (Platnick et al., 2021). For the estimation of the most
important climate parameters, i.e., the global Earth/Atmosphere radiation budget components, sensors measuring the
broadband radiation fluxes (e.g., the Clouds and the Earth's Radiant Energy System, CERES) have only been available since
1997 (Wielicki et al., 1996). The short temporal coverage and a limited measurement frequency mean that, e.g., estimations
of the global energy imbalance (EII) and its trend have considerable uncertainties which require special adjustments (Loeb et
al., 2018). However, the increasing temporal coverage of satellite-based observations and the development of new retrieval
schemes for essential climate variables (ECVs) increase the potential of using satellite-derived CDRs for environmental and
climate studies. This paper addresses this by presenting a CDR based on more than four decades of observations from polar-
orbiting meteorological satellites.

The European Organisation for the Exploitation of Meteorological Satellites (EUMETSAT) Satellite Application Facility on
Climate Monitoring (CM SAF, www.cmsaf.eu) compiles climate data records (CDRs) from various satellite sensors, mainly
aiming at describing components of the global energy and water cycle (Schulz et al., 2009). One especially important satellite
sensor for cloud and radiation studies is the AVHRR (Advanced Very High Resolution Radiometer) sensor (Cracknell, 1997,
and https://www.sciencedirect.com/topics/earth-and-planetary-sciences/advanced-very-high-resolution-radiometer). It is the
only multispectral imaging sensor aimed for meteorological observations with coverage of more than four decades until present
(2023). The AVHRR sensor first launched with the TIROS-N satellite in October 1978, and it has thereafter been carried by
numerous National Oceanographic and Atmospheric Administration (NOAA) and EUMETSAT satellites until the very last

satellite with AVHRR onboard, Metop-C, was launched in November 2018. Having nominally two satellites in sun-synchronous orbits with daytime equator crossing in the morning and the afternoon generally achieves a diurnal sampling of four global observations per day (although more frequent near the poles). CM SAF has previously compiled two editions of an AVHRR-based CDR named CLARA ("The CM SAF cLoud, Albedo and surface RAdiation dataset from AVHRR data").

The first edition, CLARA-A1 (Karlsson et al., 2013), was released in 2012 and the second edition, CLARA-A2 (Karlsson et al., 2017a), was in 2017.

The third edition, CLARA-A3, includes several extensions to CLARA-A2 regarding temporal coverage and the product portfolio. The temporal coverage is extended both backwards (with additional years 1979-1982) and forwards (with additional

years until 2020). In addition, the data record continues after 2020 with the production of a so-called Interim Climate Data Record (ICDR) based on the same algorithms as the original CDR. However, the ICDR is produced with low temporal latency (i.e., within 10 days after the end of a month), and therefore cannot use the same ancillary data or maintain the same quality in the visible radiance calibration. Nevertheless, apart from these minor differences, the data records effectively provide coverage until the present time (i.e., currently 44 years, 1979-2023) and production of the ICDR will continue until the release of the

next edition of the CLARA data record.

The most important changes in the product portfolio are the introduction of two more flavors of the surface albedo product (i.e., blue- and white-sky albedo), three additional surface radiation products (net shortwave and longwave radiation, surface radiation budget), and completely new TOA radiation budget products (i.e., reflected solar flux (RSF) and outgoing longwave

radiation (OLR)). In addition, there are also substantial algorithm improvements for products already covered by earlier CLARA editions.

This paper provides more details on the new CLARA-A3 data record, including input data, brief algorithm descriptions, product examples, and validation results. Sect. 2 briefly describes the AVHRR data record and is followed by descriptions

with separate sub-sections for the four main product groups: Cloud properties, Surface radiation, Surface albedo, and TOA radiation in section 3. Discussions follow in Sect. 4, and descriptions of data availability in Sect. 5. Section 6 provides the overall conclusions.

## 2 The historical AVHRR data record used in CLARA-A3

The AVHRR radiometer, onboard the polar-orbiting NOAA and the EUMETSAT Metop satellites, has been making

measurements since 1978. Fig. 1 gives an overview of the satellites carrying the AVHRR instrument used to produce the CLARA-A3 CDR covering the period 1979-2020. The satellites shown for 2020 in Fig. 1 are also the baseline for the ICDR processing beyond 2020, except Metop-C, which will be re-introduced at a later stage after updating to a more accurate

calibration of visible channels. The first version of the AVHRR instrument (AVHRR/1) only measured in four spectral bands. However, in 1982 a fifth channel was added (AVHRR/2), and in 1998 even a sixth channel at 1.6 µm was made available

(AVHRR/3), although it is only accessible if switched with the previous third channel at 3.7 µm.

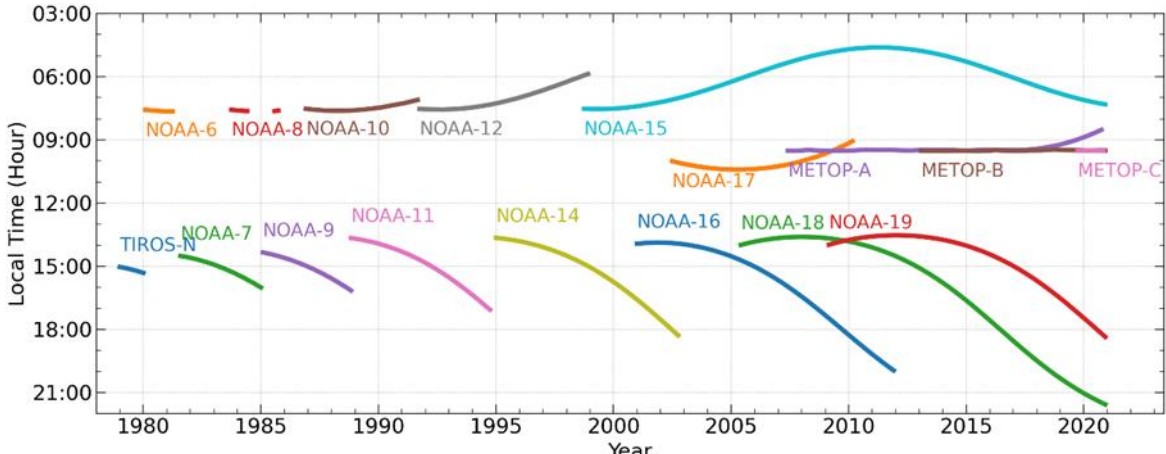

**Figure 1**: Local solar times of equator observations for all AVHRR-carrying NOAA satellites from TIROS-N to NOAA-19 and EUMETSAT's Metop A/B/C satellites. The figure shows ascending (northbound) equator crossing times for afternoon satellites (starting with TIROS-N) and descending (southbound) equator crossing times for morning satellites (starting with

NOAA-6)). Corresponding night-time observations take place 12 hours earlier/later.

Table 1 below shows further details about the AVHRR sensor and how it has changed slightly over the years, keeping most spectral channels intact. The sensor was primarily designed for operational weather and environmental monitoring, not climate monitoring. However, a system for continuously archiving a global data set with reduced 4 km resolution (Global Area

Coverage, GAC) was installed. This system has consequently enabled historical analyses of AVHRR data, including climate monitoring attempts, despite the inherent quality limitations (Karlsson et al., 2017a, Stengel et al., 2020 and Foster et al., 2022). Regarding the calibration of visible channels, quality limitations have primarily been addressed in the CLARA data record using a vicarious calibration method originally introduced by Heidinger et al. (2010). For CLARA-A3, an updated version of this method has been used (Heidinger et al., 2018) based on references to high-quality measurements from the

Moderate Resolution Imaging Spectroradiometer (MODIS), spectral band adjustments (SBAFs) from the SCanning Imaging Absorption spectroMeter for Atmospheric CartograpHY (SCIAMACHY), and by making use of Simultaneous Nadir Observations (SNOs) between individual AVHRR-carrying satellites as well as with invariant targets on Earth. For full details about the vicarious calibration method, see Heidinger et al. (2018). However, there is no inter-calibration between individual satellites for the infrared AVHRR channels. Instead, their calibration relies on standard methods utilizing reference

measurements from internal calibration targets (Kidwell, 1995 and Walton et al., 1998) for these channels.

The calibrated AVHRR reflectances and brightness temperatures, used as a basis for the CLARA-A3 data record processing, have been compiled as a stand-alone dataset in a joint CM SAF/EUMETSAT effort. This dataset, provided by EUMETSAT is denoted the AVHRR FDR (Fundamental Data Record) with the following dataset doi number: 10.15770/EUM_SEC_CLM_0060.

**Table 1**: Spectral channels of the AVHRR. The three different versions of the instrument and the corresponding satellites are described. Notice that channel 3A was only used continuously on NOAA-17 and Metop-A/B/C. The other satellites with AVHRR/3 only used the channel for shorter periods. The given wavelength ranges represent the full width at half maximum (FWHM) of the spectral response function (SRF).

| Channel Number | Wavelength (µm) AVHRR/1 TIROS-N, NOAA-6,8,10 | Wavelength (µm) AVHRR/2 NOAA-7,9,11,12,14 | Wavelength (µm) AVHRR/3 NOAA-15,16,17,18,19 Metop-A/B/C |
|---|---|---|---|
| 1 | 0.58 - 0.68 | 0.58 - 0.68 | 0.58 - 0.68 |
| 2 | 0.725 - 1.10 | 0.725 - 1.10 | 0.725 - 1.10 |
| 3A | - | - | 1.58 - 1.64 |
| 3B | 3.55 - 3.93 | 3.55 - 3.93 | 3.55 - 3.93 |
| 4 | 10.50 - 11.50 | 10.50 - 11.50 | 10.50 - 11.50 |
| 5 | Channel 4 repeated | 11.5 – 12.5 | 11.5 – 12.5 |

Additional efforts have been made to reduce or remove influence from other problematic issues in the AVHRR data record. For example, scenes or parts of scenes affected by scan motor problems and solar contamination effects are blacklisted and not used. These efforts also address sensor-specific calibration defects (other than the solar contamination issue), especially concerning channel 3B at 3.7 µm. Besides the mentioned solar contamination issue, effects like saturation in the 3.7 µm channel for hot temperatures for some early satellites and problems with unrealistic brightness temperatures near the cold extreme end of the temperature range (in particular, for satellites NOAA-15, NOAA-16, and Metop-B) have been mitigated or removed through blacklisting of parts of scenes. A special problem for the first two versions of AVHRR is the varying instrument noise in channel 3B. Therefore, a filtering technique was applied to reduce its impact on products (Karlsson et al., 2017b).

## 3 Description of CLARA-A3 product groups

Table 2 provides an overview of the product groups and all the individual products in the CLARA-A3 data record., including their abbreviations, further used in this paper. The table also gives information on the ancillary input data, product resolution, and spatial and temporal extent.

145

**Table 2:** Overview of CLARA-A3 product groups and some general characteristics concerning input data and spatial and temporal resolution. New products, not available in previous CLARA editions, are marked in bold italics. Level 2b products are daily globally sampled products for individual satellites (available exclusively for cloud products).

| CLARA-A3: CDR global clouds and radiation products+ general characteristics | | |
|---|---|---|
| Products | Cloud products | Cloud Fraction (CFC), Cloud Top Level (CTO), Cloud Phase (CPH), Liquid Water Path (LWP), Ice Water Path (IWP), Joint Cloud Histogram (JCH) |
| | Surface radiation products | Surface Incoming Shortwave Radiation (SIS), Surface Downward Longwave Radiation (SDL), *Surface Net Shortwave Radiation (SNS), Surface Net Longwave Radiation (SNL), Surface Radiation Budget (SRB)* |
| | Surface albedo products | Surface Albedo Black Sky (SAL), *Surface Albedo White Sky (WAL), Surface Albedo Blue Sky (BAL)* |
| | TOA radiation products | *Outgoing Longwave Radiation (OLR), Reflected Solar Flux (RSF)* |
| Operational satellite input | | AVHRR GAC data from NOAA and Metop satellites |
| Other operational input | | ECMWF ERA5, OSI SAF (reprocessed) ice concentrations |
| Spatial coverage | | Global (with additional polar representation for a few products) |
| Spatial resolution | | 0.25° for all products except JCH (having 1.0°) on a global grid, 25 km in two polar areas for selected products, 0.05° GAC level-2b for cloud products |
| Temporal resolution | | Daily level-2b, daily mean, pentad mean, monthly mean, histograms (depending on product, see each product description below) |
| Record length | | 1979-2020 (with ICDR extension after 2020) |

150

The following sections briefly describe all individual products from each product group. For full information on product characteristics and validation results, we refer to Algorithm Theoretical Basis Documents (ATBDs), Validation reports (VALs), and Product User Manuals (PUMs). The access link to data and documentation is available in Sect. 5.

## 3.1 Cloud properties

Cloud Fraction (CFC) is the fractional coverage (in %) of clouds within a geographic area. In level 3 (monthly mean) CLARA-A3 products, CFC is the fractional cloudiness in a 0.25° resolution grid, roughly corresponding to a 25x25 km area that decreases toward the poles. The CFC product files also provide information on the individual contributions from three vertical levels (low, medium, and high) following the definition by the International Satellite Cloud Climatology Project (ISCCP) using pressure levels 680 hPa and 440 hPa for the separation of the three vertical levels. Observe, however, that the individual contributions from low and medium levels are only the observable contributions not obscured by higher level clouds. Notice also (in Table 1) that the CFC product and all other cloud property products (except JCH) are also defined in level-2b representation with a horizontal resolution of 0.05° (approximately 5 km). The level-2b products are globally resampled images, two per day per satellite, describing ascending (passing the equator from the south) and descending (passing the equator from the north) nodes. Resampling is based on the principle that the value for the pixel with the lowest satellite zenith angle is chosen in case two or several swaths overlap. All level-3 products for cloud properties are calculated from level-2b products. All level-2b products are also available for external users. The CFC level-3 product is also prepared separately for two polar areas.

The Cloud Top Level product, CTO, describes the clouds' uppermost boundary. The product consists of three different representations:

1. Cloud Top Height, CTH
2. Cloud Top Pressure, CTP
3. Cloud Top Temperature, CTT.

CFC and CTO products are prepared for two additional areas which cover and zoom in on the polar regions. This is motivated since the standard latitude-longitude grid is not appropriate for studies focused on the polar regions because of the variable geometric grid resolution near the poles in the standard grid. The two polar regions (named South Pole and North Pole) have constant 25 km grid resolution and are used exclusively for the mentioned cloud products and for the surface albedo products (discussed in Sect. 3.3). Notice also that these Level-3 products were defined from original products per orbit (i.e., level-2 products) and not from resampled Level-2b products in order to retain as much as possible of the original information.

The thermodynamic phase product, CPH, describes whether cloud particles at cloud top level are liquid or frozen. No detection and treatment of multilayer clouds is applied in this CLARA edition. Some more detailed cloud phase information (e.g., whether liquid droplets are supercooled) is derived for the level-2 products, but this is not transferred to level-3 products.

Further cloud optical and microphysical properties are retrieved assuming that the phase from the CPH product holds for all cloud condensate in the column. This leads to two products quantifying the total integrated cloud condensate, namely liquid water path (LWP) and ice water path (IWP). For the retrieval of LWP and IWP, which is only performed during the daytime, the cloud optical thickness, COT, and the particle effective radius, CRE, are needed. COT is a measure of the attenuation of light passing through the cloud due to scattering and absorption by cloud particles. CRE is a weighted mean of the size distribution of cloud particles. Both COT and CRE are available in the product files for LWP and IWP. In addition, in the LWP product files, the cloud droplet number concentration, CNDC, and cloud geometrical thickness, CGT, are available.

There is also a hybrid product, the Joint Cloud Histogram, JCH. JCH is not a unique additional product but a histogram representation of two cloud parameters, COT and CTP, separately for both liquid and ice clouds. The product is derived since this cloud parameter representation is heavily used in the CFMIP Observation Simulator Package simulators (COSP, see Bodas-Salcedo et al., 2011) to evaluate cloud parameter simulations in climate models (e.g., in Coupled Model Intercomparison Projects, CMIPs). A COSP simulator for the CLARA dataset (Eliasson et al., 2020) has recently been updated to comply with the cloud products of the CLARA-A3 CDR. Since JCH describes the statistical distribution of COT-CTP categories, a statistically significant dataset must be available. For this reason, the grid resolution is somewhat coarser (1.0°x1.0°) than the other official level-3 cloud products. Similar Joint Cloud Histograms are provided in the ISCCP and MODIS data sets (Platnick et al. 2015).

### 3.1.1 Methods and input data

Technically, level-2 cloud parameter results were produced by the Polar Platform System (PPS) software package (https://www.nwcsaf.org/web/guest/16) developed by the Satellite Application Facility for Nowcasting applications (NWC SAF). It originates from the methods described by Dybbroe et al. (2005) for cloud masking, cloud typing and cloud top height retrievals. However, the package has since been extensively upgraded with modified algorithms and extended with cloud microphysical products (NWC SAF, 2021). The level-3 products (including the JCH product which is not included in the PPS package) were prepared and calculated by the CM SAF Operations Team at DWD.

The accurate identification of clouds is fundamental for the quality of most of the products in the CLARA-A3 data record. Cloud detection is based on Naïve Bayesian theory, where cloud probabilities for individual image feature values sequentially multiplied, followed by scaling with the likelihood of simultaneous image feature value occurrence, yielding a total cloud probability. The implemented method is called CMAPROB and is described in detail by Karlsson et al. (2020). A special feature of CMAPROB is that probabilities vary with surface type, following earlier findings by Karlsson and Håkansson (2018) based on comparisons with CALIPSO-CALIOP data (Winker, 2016). The CFC value is a binary cloud mask based on a 50% cloudiness probability threshold from CMAPROB. However, some products, primarily those that are particularly sensitive to cloud masking errors, use other CMAPROB probability thresholds (see Sect. 3.2.1 and Sect. 3.3.1). Training of the CMAPROB

method was based on global matchups between AVHRR and CALIPSO-CALIOP data for satellites NOAA-18 and NOAA-19 in the period 2006-2016.

Crucial input data for CMAPROB are the surface skin temperature, total atmospheric moisture content (i.e., column-integrated
water vapour excluding cloud water and precipitation), and snow cover from ERA5 reanalysis data (Hersbach et al., 2020). Also important is information on sea-ice occurrence taken from re-processed ice concentration datasets from the EUMETSAT SAF on Ocean and Sea-Ice (OSI SAF project (https://osi-saf.eumetsat.int/, OSI-450 (DOI: 10.15770/EUM_SAF_OSI_0008)) combined with the operational extension of the data record OSI 430-b (DOI: 10.15770/EUM_SAF_OSI_NRT_2008). For the ICDR processing the near-real-time processed parameters OSI- 401b (DOI: 10.15770/EUM_SAF_OSI_NRT_2004) and OSI-
401d have been used. CMAPROB also uses monthly mean land surface emissivities derived from MODIS data (https://modis.gsfc.nasa.gov/data/dataprod/mod11.php).

CFC uncertainties for the Level 2b product can be interpreted directly from the CMAPROB product which is provided together with the binary cloud mask. Maximum uncertainty is found at the 50% cloud probability level. For the level-3 product, a
simple estimation based on the averaging of the probability distance from the 50% threshold for clear and cloudy pixels is provided.

The CTO product is based on an artificial neural network (ANN, of the type multilayer perceptron) trained offline with collocations of passive imager measurements from AVHRR and CALIPSO-CALIOP (Winker et al., 2016) cloud top pressure
observations (as outlined by Håkansson et al., 2018). The 11 µm channel is used together with the 12 µm channel for AVHRR/2 and AVHRR/3. However, when CTO is based on AVHRR/1, where the 12 µm channel is missing, the 3.7 µm channel is used instead. The CTO product also includes error estimates retrieved using Quantile Regression Neural Networks (see Pfreundschuh et al., 2018) for the 16th and 84th percentiles.

For estimating the CTO uncertainty in the level-2b product, the absolute CTO difference from the 16th and the 84th percentile is provided. These CTO 1-sigma uncertainties from Level-2b files are then propagated into level-3 products, following Stengel et al. (2017).

To determine CPH, a series of spectral tests are applied as described in Pavolonis et al. (2005). The first step is to assign the
cloudy measurement to one of five cloud types, liquid, supercooled, opaque ice, cirrus, or overlap. Based on the cloud type, the cloud is retrieved as liquid phase (the first two) or ice phase (the last three). The cloud phase is changed if it is inconsistent with the retrieved cloud top temperature, i.e., when liquid phase is retrieved for CTT < 231 K or when ice is retrieved for CTT > 265 K. Selection of these thresholds was motivated based on the temperature limits where liquid droplets and ice crystals occur in supercooled clouds, which are included in the extended cloud phase retrieval. These numbers do not match exactly

respective limits given in the literature: -40 ºC (e.g. Tabazadeh et al. 2003) and -6 ºC (e.g. Hobbs and Rangno, 1985). For the CPP retrieval algorithm, the specific limits were selected empirically based on comparisons with observations from Cloudsat and CALIPSO. Finally, the CPH product is provided as the fraction of liquid clouds.

COT and CRE are retrieved using the classical Nakajima and King (1990) approach, based on the principle that the reflectance of clouds at a non-absorbing (for cloud particles) visible wavelength (here 0.6 µm) is strongly related to COT but has little dependence on CRE. In contrast, cloud reflectance at an absorbing wavelength in the shortwave-infrared region (here 1.6 or 3.7 µm) strongly depends on CRE. The TOA reflectance of homogeneous, plane-parallel clouds at these wavelengths is simulated with the doubling-adding KNMI (DAK, De Haan et al., 1987; Stammes, 2001) radiative transfer model for a range of clouds and viewing/illumination geometries and stored in a look-up table (LUT). COT and CRE are then retrieved for the
assigned phase by iteratively matching the simulated with the observed reflectances. These two parameters are finally used to compute LWP and IWP, for liquid and ice clouds, respectively, as in Stephens (1978). Since the retrieval of COT and CRE requires information from visible AVHRR channels, these products are only determined for daytime conditions (currently defined by solar zenith angles below 75°).

Estimated uncertainties in reflectance measurements and various input variables (e.g., surface albedo, total ozone column) are propagated to yield uncertainty estimates in retrieved COT and CRE. These are, in turn, propagated to uncertainty estimates in LWP, IWP, CDNC and CGT. Further details are given in NWC SAF (2021). Notice, however, that uncertainties do not include deviations from the assumption of horizontal and vertical homogeneity of the clouds as a source of error.

The main algorithm updates for COT, CRE, LWP, and IWP compared to CLARA-A2 are:
1. Improved uncertainty estimates of the retrieval products taking into account a more comprehensive range of error sources. This has made the uncertainty estimates more realistic (although that is very hard to prove) and it has given insight into the relative importance of error sources, but the overall magnitude of the uncertainty estimates has not changed much compared to CLARA-A2.
2. Revised radiative transfer simulations based on narrower liquid droplet size distributions and an ice crystal model of severely roughened aggregated solid columns (Yang et al., 2013 and Baum et al., 2011)
3. Additional retrieval of CDNC and CGT for liquid clouds, following Bennartz and Rausch (2017), if channel 3b (at 3.7 µm) is available for the retrieval.

### 3.1.2 Product examples and validation results

To illustrate how the CLARA-A3 CFC parameter performs on the global scale, Fig. 2 shows a time series of monthly mean global CFC compared with six other reference datasets, including results for the predecessor CLARA-A2. Notice that 3 1/2 years of data have been added to the original CLARA-A2 CDR. This extension is called CLARA-A2.1 (with DOI page

http://dx.doi.org/10.5676/EUM_SAF_CM/CLARA_AVHRR/V002_01). Two of the reference datasets, ESA-CCI (Stengel et al., 2020) and PATMOS-x (Foster et al., 2022), are also based on AVHRR data. However, the latter uses sounding data from the High-resolution Infrared Radiation Sounder (HIRS) in addition to AVHRR data. A third reference dataset is based on MODIS data (Platnick et al., 2015), and a fourth is on combined geostationary and polar data (ISCCP-HGM, Young et al., 2018). The remaining two references are based on cloud information from the cloud lidar CALIOP onboard the CALIPSO satellite (CALIPSO Science Team, 2021). The CALIPSO-based data record is a product compiled for the Global Energy and Water Cycle Experiment (GEWEX) and prepared in two flavours: Passive and Top Layer (TL). The latter flavour includes all CALIOP-detected clouds, while the passive flavour includes all clouds except clouds having integrated cloud optical thicknesses smaller than 0.3.

We notice in Fig. 2 that CLARA-A3 CFC has generally increased by approximately 3 % in absolute values compared to CLARA-A2. The detailed validation based on CALIOP and surface (SYNOP) observations confirmed this improvement. The largest improvements are found over the Arctic Ocean and for mid-latitudes over the Southern Ocean (Karlsson et al., 2023a). Conditions over Antarctica are still challenging during polar winter conditions when a large fraction of all clouds is not detected. However, it is worth mentioning that cloud detection over snow- and ice-cover during illuminated conditions in the polar summer works well over both poles. Compared to the complete CALIOP dataset (i.e., including all CALIOP-detected clouds), the bias has decreased from -15.1 % for CLARA-A2 to -11.1 % for CLARA-A3. Compared to SYNOP, the bias has changed from -3.1 % for CLARA-A2 to 2.0 % for CLARA-A3. It can also be concluded from Fig. 2 that CLARA-A3 CFC values are now confined in between the values of CALIPSO-TL and CALIPSO-Passive, while CLARA-A2 results are lower than the CALIPSO-Passive results. This indicates that a substantial fraction of all clouds with an optical thickness less than 0.3 are now detected in CLARA-A3. This clearly differs from the performance of the predecessor CLARA-A2 where less thin clouds were detected.

Global CFC levels appear relatively constant, with a slight negative trend. Only ISCCP-HGM indicates a larger negative trend than -1 % per decade. CLARA-A3 is in best agreement with the MODIS and PATMOS-x datasets. An interesting observation from Fig. 2 is that results from the latest versions of the different data records have converged, i.e., showing better agreement with each other than previous inter-comparison studies have shown (Stubenrauch et al., 2013, and Karlsson and Devasthale, 2018). For more details and better visibility of all results in Fig.2, we refer to Karlsson et al. (2023a + 2023b).

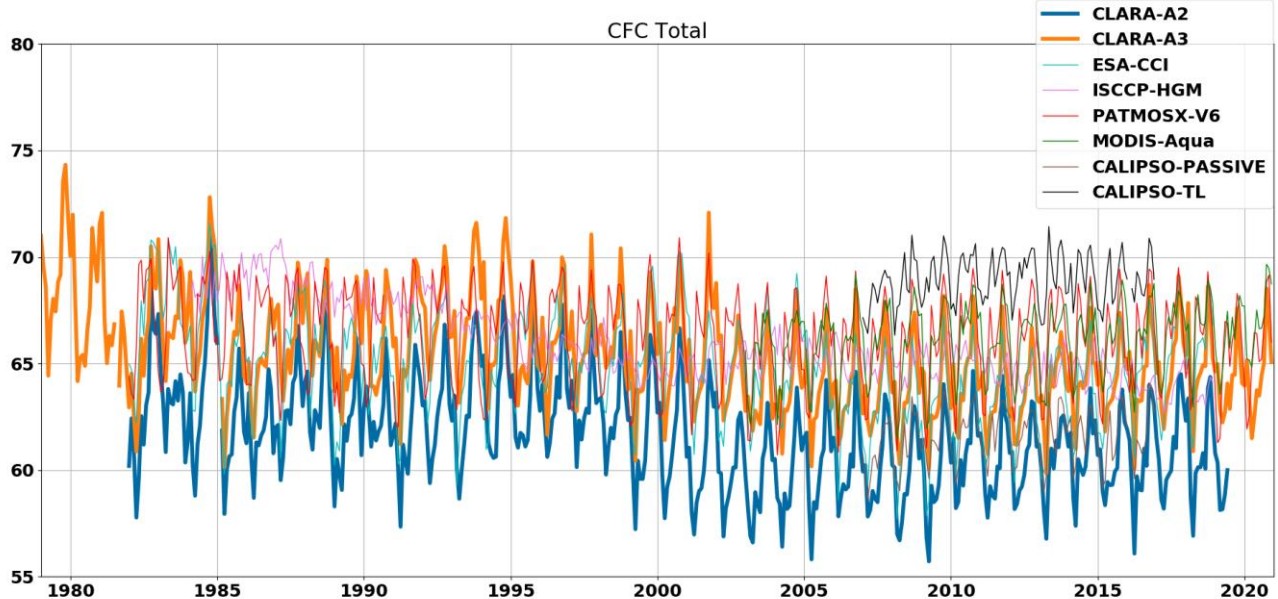

**Figure 2:** Monthly mean global CFC for CLARA-A3 (thick orange curve) compared with the previous CLARA-A2 data record (thick blue curve) from 1979-2020. The remaining curves represent other recent data records based on AVHRR data and other polar and geostationary data (see text for more details).


Fig. 3 shows the global mean cloud top pressure (CTP) for the same CDRs as in Fig. 2. We notice a much larger spread of results among the data records for CTP compared to CFC in Fig. 2. However, there are no obvious trends for any data record (however, see the discussion of CLARA-A3 results for the first two satellites TIROS-N and NOAA-6 in Sect. 4). Thus, it appears that mean global cloud tops are relatively stable and indifferent. CLARA-A3 generally retrieves 50-75 hPa lower (i.e.,
at least 1000 m higher) cloud tops than CLARA-A2. This is a substantial improvement, as deduced from the detailed CALIOP validation and the fact that results are now safely confined in between the two flavours of the CALIPSO dataset. Another remarkable feature is that CLARA-A3 results appear even closer to the CALIPSO-TL results than other datasets (e.g., PATMOS-x).

An in-depth study of deseasonalized anomalies and trends of both cloud amounts and cloud top heights for CLARA-A3 and
other similar CDRs can be found in Devasthale and Karlsson (2023).

Validation results based on the CALIPSO-TL results show a mean CTP bias of 27 hPa. This is a substantial improvement compared to the corresponding results of the CLARA-A2 CDR and, clearly, the best result among all of the investigated CDRs in Fig. 3. For full details on the validation results, see the Clouds VAL report (accessible with the link given in Sect. 5).


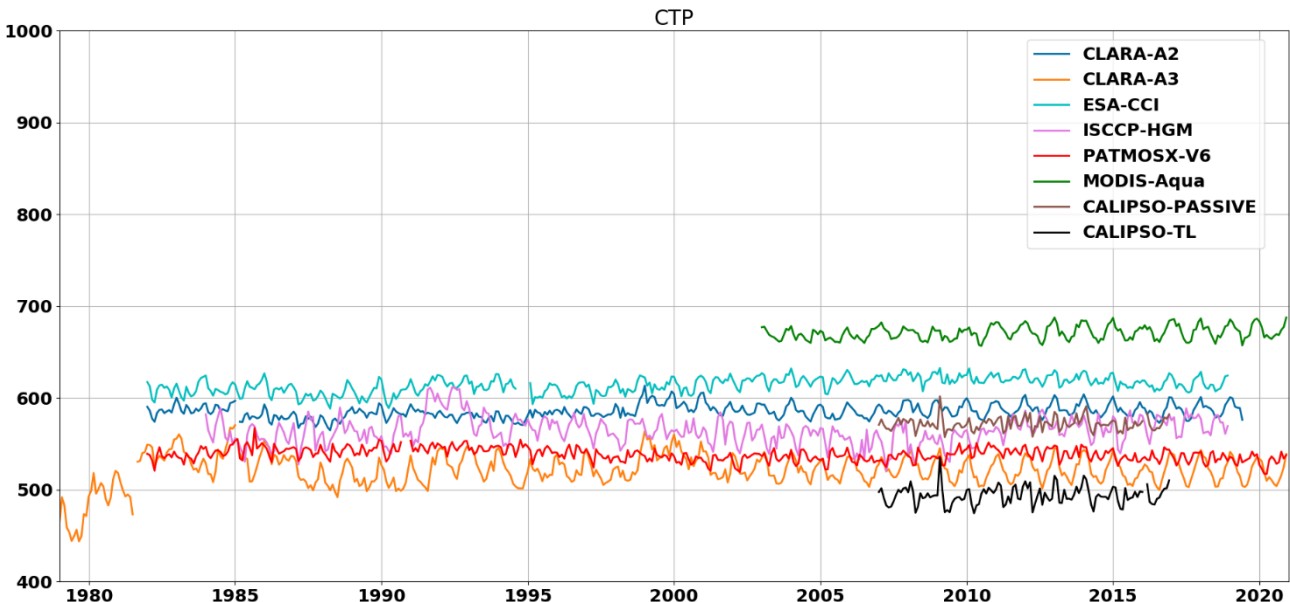

**Figure 3:** The monthly mean global cloud top pressure (CTP in hPa) for CLARA-A3 (thin orange curve) compared with the same data records shown in Fig. 2.

Fig. 4 compares the global average of CPH with the ESA-CCI (Stengel et al., 2020) and MODIS-Aqua (Platnick et al., 2017)

products. All datasets agree reasonably well in terms of seasonality. The lower CPH values of CLARA-A3 compared to CLARA-A2 should be largely attributed to differences caused by the new, improved CTH retrieval algorithm: higher (ice) clouds are detected in the new CLARA edition, causing a decrease in the liquid cloud fraction. The CPH time series looks rather stable except for the first years when TIROS-N and NOAA-6 were active (see also Fig. 3). Validation with CALIPSO level-2 data suggests that CLARA-A3 liquid cloud fraction has a bias of about -2 % (i.e., a small underestimation). By

comparison, the mean differences with MODIS and ESA-CCI are -5 % and -4 %, respectively.

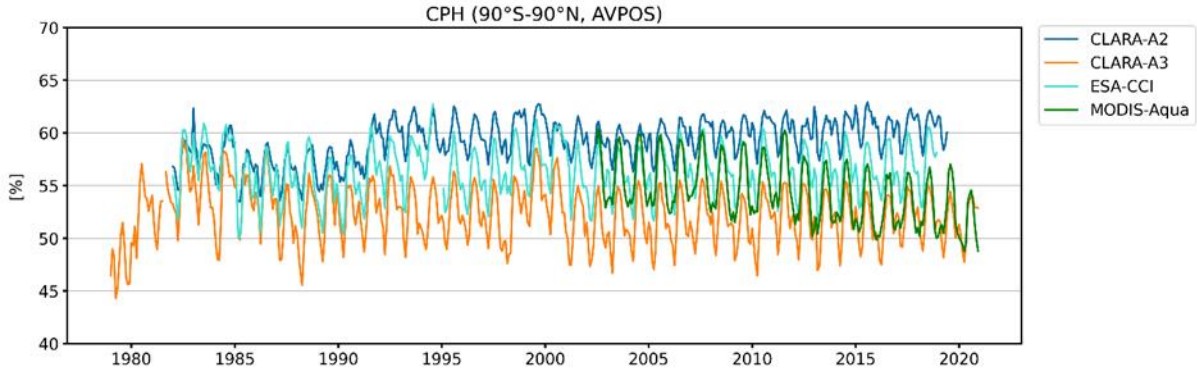

**Figure 4**: Time series of global monthly mean liquid cloud fraction (i.e., liquid/(liquid+ice), in %) from CLARA-A3, CLARA-A2, ESA-CCI, and MODIS-Aqua.

Spatial distributions of all-sky LWP and IWP are shown in Fig. 5. High latitudes are excluded from these multi-year (2003-2016) averages due to illumination conditions prohibiting retrievals during local winter. The all-sky LWP from CLARA-A3 is generally lower than the reference data sets (Fig. 5a), partly due to a lower liquid cloud fraction overall. The main spatial features are the somewhat larger differences in the southern latitudes, where absolute values are also large (apparent especially compared to MODIS and ESA-CCI), and the positive differences in Greenland and parts of Canada. The latter is probably due

to snow or ice on the surface, which complicates the retrievals. In the all-sky IWP spatial features (Fig. 5b), the Inter-Tropical Convergence Zone (ITCZ), where ice clouds prevail, is highlighted. At the same time, very low values occur over large oceanic regions with low (liquid) stratocumulus clouds. Overall CLARA-A3 values are higher over land and lower over ocean compared to MODIS and lower in general compared to ESA-CCI. Mean differences between CLARA-A3 and MODIS amount to -6 and 0 g m$^{-2}$ for LWP and IWP, respectively, while mean differences with ESA-CCI are -9 and -15 g m$^{-2}$ for LWP and

IWP, respectively.

Additional details on validation results are provided in the dedicated CLARA-A3 Cloud Products validation report available through the link given in Sect. 5.

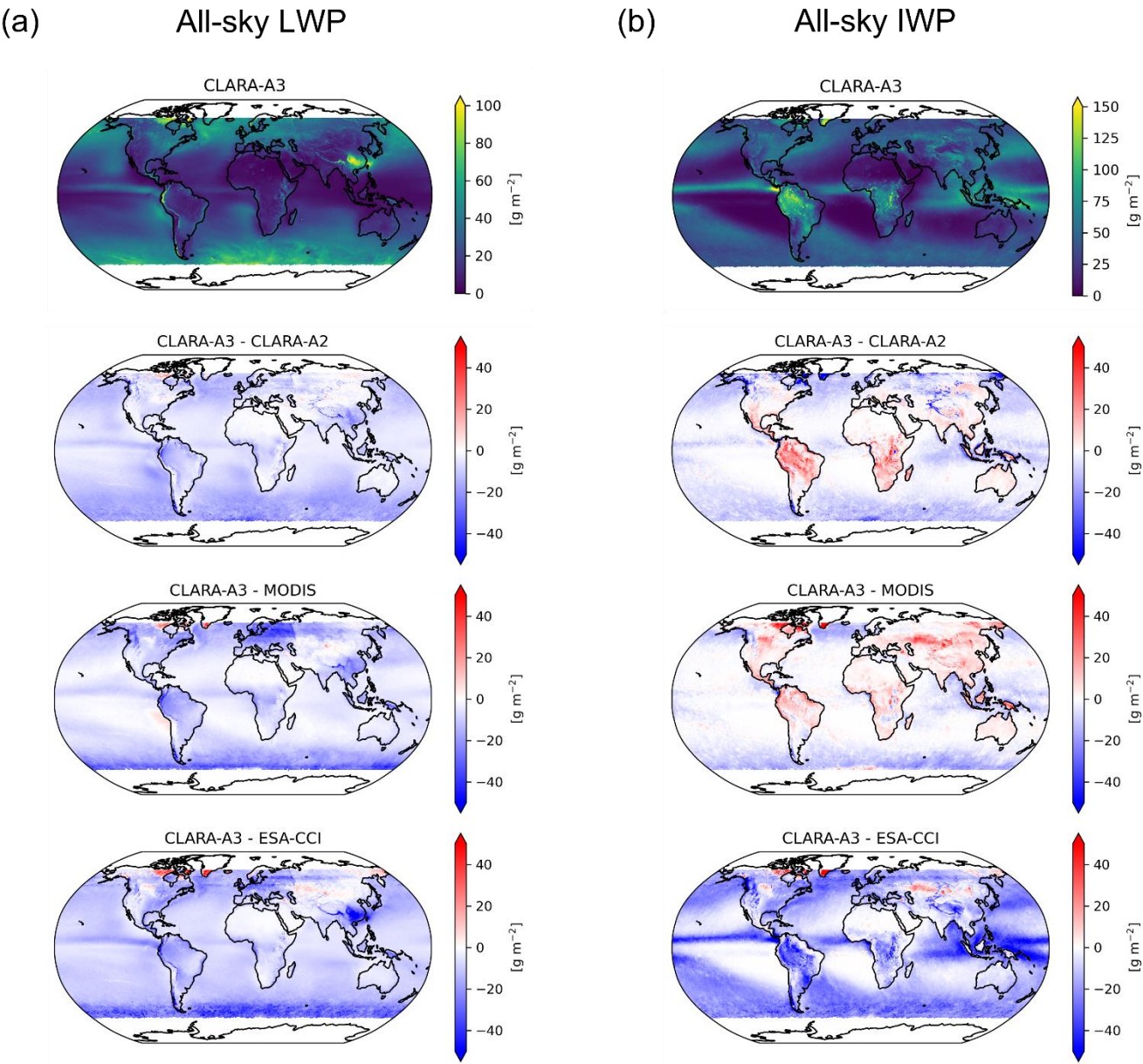

**Figure 5:** Average all-sky liquid (a) and ice water path (b) from CLARA-A3 (top left panels in a and b) and differences with respect to CLARA-A2, MODIS-Aqua, and ESA-CCI, calculated from 2003 to 2016.

### 3.2 Surface radiation

The CLARA-A3 CDR includes the downward and net components of the shortwave and longwave radiation and the total surface radiation budget (see Table 2). Also described below is the method to calculate the upward components (needed to compute the net components).

### 375 3.2.1 Methods and input data

The retrieval of the surface solar radiation products closely ties with the satellite measurements from the AVHRR instruments. In contrast, the retrieval of surface longwave radiation requires substantial auxiliary information due to the fundamental limitations of satellite-based imaging sensors.

### 380 Shortwave Radiation

The basic method to estimate the surface solar irradiance, i.e., the downward solar radiation, in CLARA-A3 has remained the same compared to previous editions of the CLARA data record (Mueller et al., 2009; Karlsson et al., 2013). The changes and improvements in the data quality of the CLARA-A3 surface solar radiation data record (compared to CLARA-A2) can be attributed to the use of improved input data. The method now uses the information from the probabilistic cloud mask (see

Section 3.1.1) to classify each pixel as clear-sky or cloudy. Furthermore, the reflected solar radiation flux, which had been estimated with a very basic method in CLARA-A2, is now being derived with a more advanced algorithm. Finally, the use of surface albedo from ERA-5 (compared to a climatology used in CLARA-A2) also improves the surface radiation estimation for cloudy and clear-sky pixels.

The instantaneous surface irradiance, $I_i$, for pixels identified as clear-sky is estimated using the Mesoscale Atmospheric Global Irradiance Code (MAGIC) clear-sky model (Mueller et al., 2004). For satellite pixels identified as cloudy, a look-up-table-approach is applied to relate the reflected solar flux (as derived in the CLARA-A3 processing of TOA radiation fluxes, see Section 3.4) to the atmospheric transmissivity, which is then used to estimate pixel-level surface irradiance, $I_i$. The threshold used in the probabilistic cloud mask to identify cloudy pixels is set to 50 % for most surface types but adjusted to 90 % for

bright surfaces to enhance the confidence in the cloud detection under these conditions (a cloud-conservative approach). For each satellite pixel, even if determined to be cloudy, the corresponding clear sky irradiance, $I^{clr}$, is estimated. It is important to note that the satellite retrieval of surface irradiance is limited to solar zenith angles below 80° due to limitations in the look-up table for cloudy situations. The surface irradiance is not retrieved at larger solar zenith angles, i.e., low sun situations.

Auxiliary data are required to calculate the instantaneous clear-sky surface irradiance and prepare the look-up table (LUT) that relates the reflected shortwave flux to the transmissivity. These auxiliary data include surface albedo, atmospheric water

vapour, and aerosol optical properties. For surface albedo, daily data from the ERA5 reanalysis (Hersbach et al., 2020) are used for the CDR and ERA5T (i.e., a preliminary version of ERA5, see Sect. 3.5 for further explanation) for the ICDR, respectively. Similarly, ERA5/ERA5T provides the vertically-integrated atmospheric water vapour. In particular, using temporally-varying surface albedo data substantially improves the surface albedo compared to previous versions of the CLARA data record. The aerosol optical properties used for the pixel-level retrieval are unchanged compared to previous editions, i.e., based on climatological information (see Karlsson et al., 2013).

The daily averaged surface irradiance, $I_{dm}$, is estimated using the instantaneous retrieval results, $I_i$, using all observations from the corresponding day (in UTC) by Eq. (1) (based on Diekmann et al., 1988):

$$I_{\mathrm{dm}} = \frac{I_{\mathrm{clr,dm}} * \sum I_i}{\sum I_i^{\mathrm{clr}}} \tag{1}$$

Here, $I_{clr,dm}$ is the clear sky daily mean derived from clear-sky model simulations using the MAGIC clear sky model. By weighting the clear-sky daily mean surface irradiance with the ratio of the sum of the all-sky to the sum of the clear-sky irradiance, this formula allows the estimation of daily averaged surface irradiance at high accuracy with few observations. The resulting accuracy depends on the diurnal variability of cloud coverage, which is better observed with more and properly temporally-spaced observations.

The accuracy of the daily averaged all-sky surface irradiance is also determined by the estimation of the daily clear-sky irradiance. The aerosol information used as an input parameter for the daily-mean clear sky irradiance differs from the one used in the estimation for the instantaneous irradiance. Monthly information about the tropospheric aerosol optical depth and their physical properties are taken from model-based estimates (Fiedler et al., 2019 a, b). These estimates are based on assumptions about the pre-industrial natural aerosol and emission inventories (Fiedler et al., 2019a) as well as on emission scenarios (Fiedler et al., 2019b). For the generation of CLARA-A3, the monthly climatology MACv2 of natural aerosol data has been used (Fiedler et al., 2019a); the anthropogenic aerosol is prescribed by MACv2-SP (1979 – 2014), and by the scenario SSP2-45 (2015 - 2025) (Fiedler et al., 2019a, b). These aerosol scenarios are also part of the Scenario Model Intercomparison Project (ScenarioMIP) of CMIP6. Due to the remaining uncertainty of the long-term variability and trend of the monthly aerosol data, the mentioned aerosol data have been used to generate a multi-year monthly climatology, which has been used to estimate the daily clear-sky surface solar irradiance. Hence, the clear-sky surface irradiance does not include any year-to-year aerosol variability or long-term aerosol trend. Noteworthy is also that day-to-day aerosol variability of aerosols can be substantial and is not compensated for here. However, despite this inability to in detail describe the temporal evolution of aerosols, we regard the impact of aerosols to be marginal (i.e., a few % uncertainty of the surface irradiance) in comparison to other factors, where cloudiness is the dominating one.

The daily mean surface radiation on the final 0.25°x0.25°-grid is obtained by averaging the corresponding 25 high-resolution (i.e., 0.05°x0.05°) grid boxes. However, if based on fewer than 20 observations, the daily mean is set to the missing value. When estimating the daily average surface irradiance, no limit is set on the number of available satellites or satellite overpasses. This implies that the daily average might be calculated from only one satellite overpass, assuming 20 or more valid satellite pixels are available in the 0.25°x0.25° grid box. In this case, only the spatial variability of the daily means is considered during the averaging. The accuracy of the gridded daily mean is expected to be substantially reduced under these conditions, particularly in regions with a pronounced diurnal cycle of cloud coverage, which can only be partially observed by a single satellite. In addition, surface irradiance values in all grid boxes during polar night are set to zero. The monthly means of the surface solar irradiance are calculated as averages from the daily mean values. However, grid boxes with 20 or fewer valid daily means of surface solar radiation are considered missing data.

Note that the requirements on data availability for the estimation of daily mean surface irradiance, particularly in combination with the 80°-threshold of the solar zenith angle for grid boxes in polar twilight regions, result in missing data in the daily averaged data and, subsequently, also in the monthly-averaged data. The grid boxes with systematically missing data prevent a straightforward estimation of globally averaged surface irradiance and the surface net solar radiation..

The surface net solar radiation from the CLARA-A3 data record are only monthly averages based on the daily averages of surface irradiance and the surface-reflected radiance computed by use of the pentad-averaged blue-sky surface albedo provided as part of the CLARA-A3 data record (see Section 3.3). The monthly averaged surface net shortwave radiation is derived from the daily mean values requiring a minimum of 20 valid daily mean data each month.

**Longwave Radiation**

The surface downwelling longwave radiation is determined by the properties of the lowest atmospheric levels, i.e., the temperature, the humidity, and the cloud base height (if clouds are present, see Ohmura, 2001). Unfortunately, there is little or no information on these properties in visible and infrared satellite measurements, such as those from the AVHRR instrument (Ellingson, 1995). This prevents the estimation of surface longwave radiation from these satellite measurements without additional data sources, which will then provide a substantial part of the information. The surface longwave radiation in the CLARA-A3 data record is only estimated and provided as monthly averages. These data are closely linked to the corresponding estimates from ERA-5.

The spatial resolution of the global reanalysis data is comparable with the CLARA-A3 spatial grid resolution and we can therefore directly map the ERA5T data to the global 0.25°x0.25°-grid of the CLARA-A3 data record. To enhance the consistency between the surface downward longwave radiation and the other CLARA-A3 products, a small local adjustment was applied to the surface downwelling longwave data using the monthly mean cloud fraction data from CLARA-A3 and an updated cloud correction factor (Karlsson et al., 2013). The monthly mean upward longwave surface radiation, necessary to

derive the net longwave surface radiation, is taken directly from ERA5T. Finally, the monthly averaged surface radiation
budget is the sum of the net shortwave and net longwave surface radiation.

### 3.2.2 Product examples and validation results

The quality of the CLARA-A3 surface radiation data records has been determined by comparison with surface reference
measurements from the Baseline Surface Radiation Network (BSRN; Driemel et al., 2018) network. The overall numbers of
these comparisons are listed in Tab. 3.

**Table 3:** Global averages of the CLARA surface radiation parameters, the number of observations from the BSRN network
used in the validation as well as the results of the validation, namely the mean difference, the mean absolute difference, and
the correlation of the monthly anomalies. The two lines for SIS correspond to the validation results of the monthly mean (mm)
and the daily mean (dm) data. The number of observations refers to the absolute number of available monthly / daily mean
reference data used in the validation.

| Data Set | Global Mean, W/m² | Number of Obs | Mean Diff, W/m2 | Mean Abs. Diff, W/m² | Anomaly Corr |
|---|---|---|---|---|---|
| SIS | 185.5 | | | | |
| mm | | 9 369 | 1.9 | 7.3 | 0.91 |
| dm | | 262 280 | 1.6 | 16.9 | |
| SDL | 340.8 | 9 530 | −5.8 | 7.2 | 0.90 |
| SNS | 163.2 | 2 165 | 9.5 | 10.8 | 0.89 |
| SNL | −56.6 | 2 363 | −4.3 | 6.8 | 0.84 |
| SRB | 107.2 | 2 072 | 5.4 | 9.7 | 0.64 |

The global average of the surface radiation data records is derived using additional information from CERES-SYN1deg (Rutan
et al., 2015), V4.1. As mentioned earlier, the missing data in surface irradiance along the twilight regions prevent a meaningful
calculation of global averages of the shortwave radiation components. To account for these missing data, CERES-SYN1deg
data from 2001 to 2020 are used to assess the impact of these missing grid values on the long-term averages and apply the
corresponding correction term to the CLARA-A3 multi-year monthly averages.

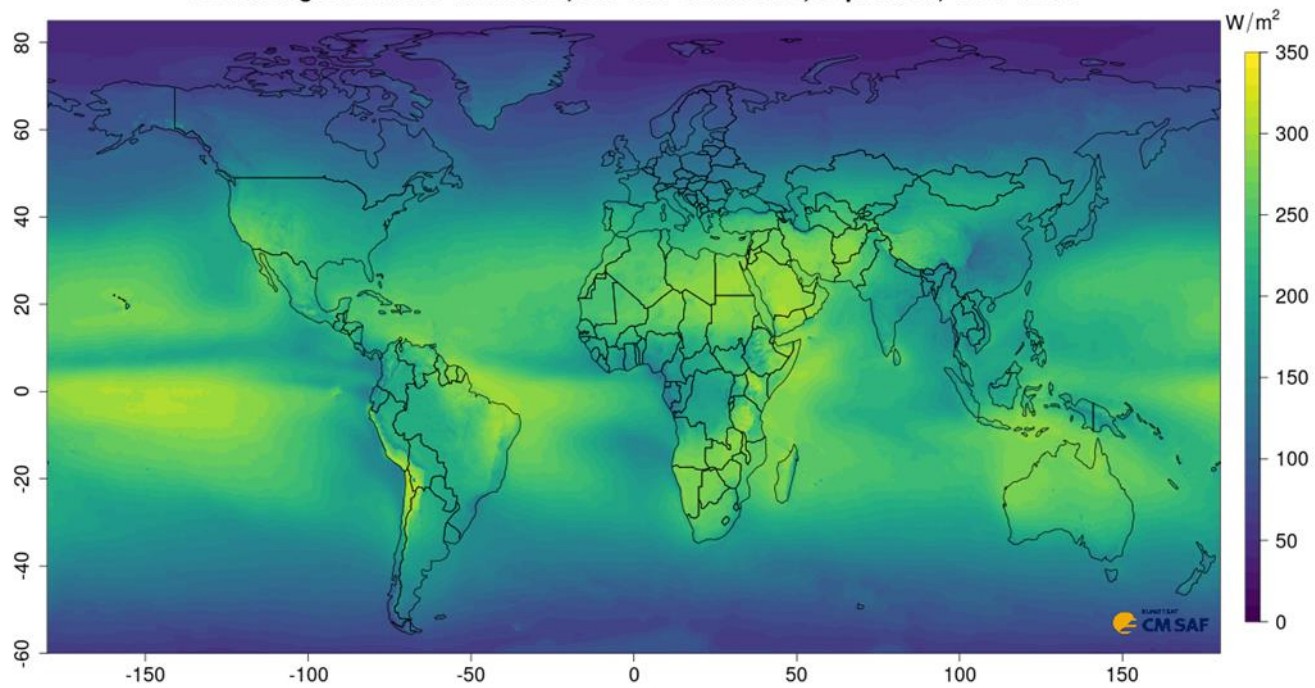

**Figure 6:** Average surface irradiance for September (1991 to 2020) based on the CLARA-A3 SIS data record.

Figure 6 shows the spatial distribution of the climatological surface irradiance for September, averaged over 1991-2020. Evidently, the CM SAF CLARA-A3 data record allows, for the first time, the estimation of climatological averages for an official 30-year WMO reference period, namely from 1991 to 2020.

More details on validation results can be found in the detailed CM SAF validation report (available through the link in Sect. 5).

### 3.3 Surface albedo

The surface albedo (hemispherical reflectance) of the Earth's surface depends not only on the optical properties of the surface but also on the directionality of the incoming solar radiation. Therefore, surface albedo is commonly quantified in three 'flavors': the directional-hemispherical reflectance (or black-sky albedo), the bidirectional reflectance under fully diffuse illumination (or white-sky albedo), and the bidirectional reflectance under ambient illumination (or blue-sky albedo). In CLARA-A3, estimates for all these parameters are provided for the first time, while the predecessor CLARA records only contain the black-sky albedo. The black/white/blue-sky albedo variables are denoted with SAL, WAL, and BAL, respectively. All albedo estimates are provided in the standard CLARA spatial grid resolution of 0.25°x0.25°, with the polar regions covered

with 25 kmx25-km resolution subsets in the EASE-2 projection. Temporally, the data are provided as five-day (pentad) and monthly means. The data are not normalized to any specific solar geometry (Sun zenith angle). However, the corresponding data are provided with the albedo estimates for users wishing to make such normalizations themselves.

### 3.3.1 Methods and input data

A complete description of the retrieval process is available in the Algorithm Theoretical Basis Document (ATBD) of CLARA-A3 surface albedo products and the associated reference publications, available through the CLARA-A3 DOI link (see section 5). In general, the retrieval seeks to first estimate the black-sky albedo (SAL) from the intercalibrated AVHRR observations and supporting data and then estimate the white- and blue-sky albedos from that through empirical relationships. The SAL algorithm proceeds sequentially from topography and atmospheric corrections to the treatment of angular reflectance isotropy,
the derivation of spectral albedos, and finally, the estimation of broadband surface albedo. Snow and ice are identified during processing and treated differently from land surfaces.

The identification of clear-sky areas in each AVHRR overpass is followed by an inversion of clear-sky surface reflectance for AVHRR channels 1 and 2 from the corresponding TOA reflectances. A key feature of the CLARA-A3 record for this purpose
is the novel availability of a probabilistic cloud mask; this data is used for both cloud screening of individual AVHRR overpasses (all observations with cloud probability >20% are discarded) as well as in the aggregation of temporal means of surface albedo (Manninen et al., 2022). The topography correction for geolocation and radiometry of the AVHRR observations is carried out equivalently to the preceding CLARA albedo records (Manninen et al., 2011). All observations with an unfavourable observation geometry (Sun Zenith Angle >70 deg. or Viewing Zenith Angle >60 deg.) are discarded.

The atmospheric correction necessary to obtain directional surface reflectances is again based on the Simplified Method for Atmospheric Correction (SMAC; Rahman and Dedieu, 1994). The data source for atmospheric composition (water vapour, ozone, surface pressure) is now the ERA5 reanalysis for CDR and ERA5T for the ICDR. To account for the aerosol loading of the atmosphere, we use the same observation-based aerosol optical depth (AOD) data record as for the CLARA-A2
predecessor records (Jääskeläinen et al., 2017). The last years of the CLARA CDR record and the following ICDR are processed with an aerosol climatology (mean of years 2005-2014) instead of yearly and monthly data. This is due to caution related to recent observations of some degradation in the UV channel calibrations of the OMI sensor, our data source for AOD in this period (Kleipool et al., 2022).

Treating bidirectional reflectance effects (i.e., BRDF) is equivalent to preceding CLARA albedo records, as is the narrow-to-broadband conversion (NTBC) to surface broadband albedo. As before, we do not attempt to correct for BRDF effects in the reflectance of snow and ice in the overpass processing. Instead we rely on sufficiently dense temporal and angular sampling to simply average the surface reflectances into a realistic estimate for albedo. It should be kept in mind that the NTBC used

for snow and ice (Xiong et al., 2002) also self-adapts to wet and dry surfaces, improving accuracy for the wide range of possible cryospheric surface conditions.

The most novel aspect of CLARA-A3 surface albedo record is the availability of white- and blue-sky surface albedo estimates (WAL and BAL). WAL over snow-free land surfaces is estimated following Yang et al. (2008). Over snow and ice, WAL estimation is based on statistical relationships observed in in situ albedo measurements (Manninen et al., 2019), with adaptations between open and forested landscapes. BAL is derived as the direct irradiance fraction-weighted mean of SAL and WAL (e.g., Pinty et al., 2005). The direct irradiance fraction is derived with a statistical relationship that links cloud probability (CMAPROB) with the clearness index (Hofmann and Seckmeyer, 2017). Snow-free and snow/ice AVHRR observations are combined in the final temporal averages by weighting their respective observation counts.

The albedo data do not contain uncertainty estimates per grid cell. However, a wide variety of parameters describing the statistical distribution and sampling density of the retrieved albedos are provided in the data files, e.g. skewness, kurtosis, and number of valid observations per grid cell.

### 3.3.2 Product examples and validation results

The new global blue-sky albedo (BAL) estimates of CLARA-A3 are illustrated in Fig. 7, along with the locations of reference sites. Three principal reference data sources were used: first was the BSRN for land surfaces, limited to those sites with measurements of reflected solar radiation and non-coastal locations (N=13). The second is the Programme for the Monitoring of the Greenland Ice Sheet (PROMICE; Fausto et al., 2021), whose automated stations provide snow/ice albedo measurements across the ice sheet (N=25). Finally, for (Arctic) sea ice, we employed in situ measurements of surface albedo from the SHEBA (Perovich et al., 2002) and Tara Arctic (Vihma et al., 2008) ice camps. Data gaps typically occur from either invalid solar illumination geometry (Antarctica in the example) or persistently too-high aerosol loading of the atmosphere (Siberia and Eastern China in the example). In the pentad means, persistent cloudiness may also cause transient data gaps.

Circles overlaid on Fig. 7 show the overall mean (relative) bias of CLARA-A3 BAL against in situ observations from the BSRN and PROMICE networks. Observed biases are generally low, excepting some sites where terrain heterogeneity is high from the albedo perspective, such as sites near edges of ice sheets or mixed forest-grassland regions. Geographical coverage of validation sites strongly favours the Northern Hemisphere, as only a limited subset of the global BSRN network provides measurements of reflected solar radiation necessary for albedo derivation. Also, while Fig. 7 provides an overview of CLARA-A3 BAL bias against in situ reference measurements, the intent here is not to provide details on the validation process or its results. The full description of the evaluation is available either from the Validation Report (VAL) or the associated scientific papers on CLARA-A3 surface albedo, available through the DOI site of the data record (given in Sect. 5).

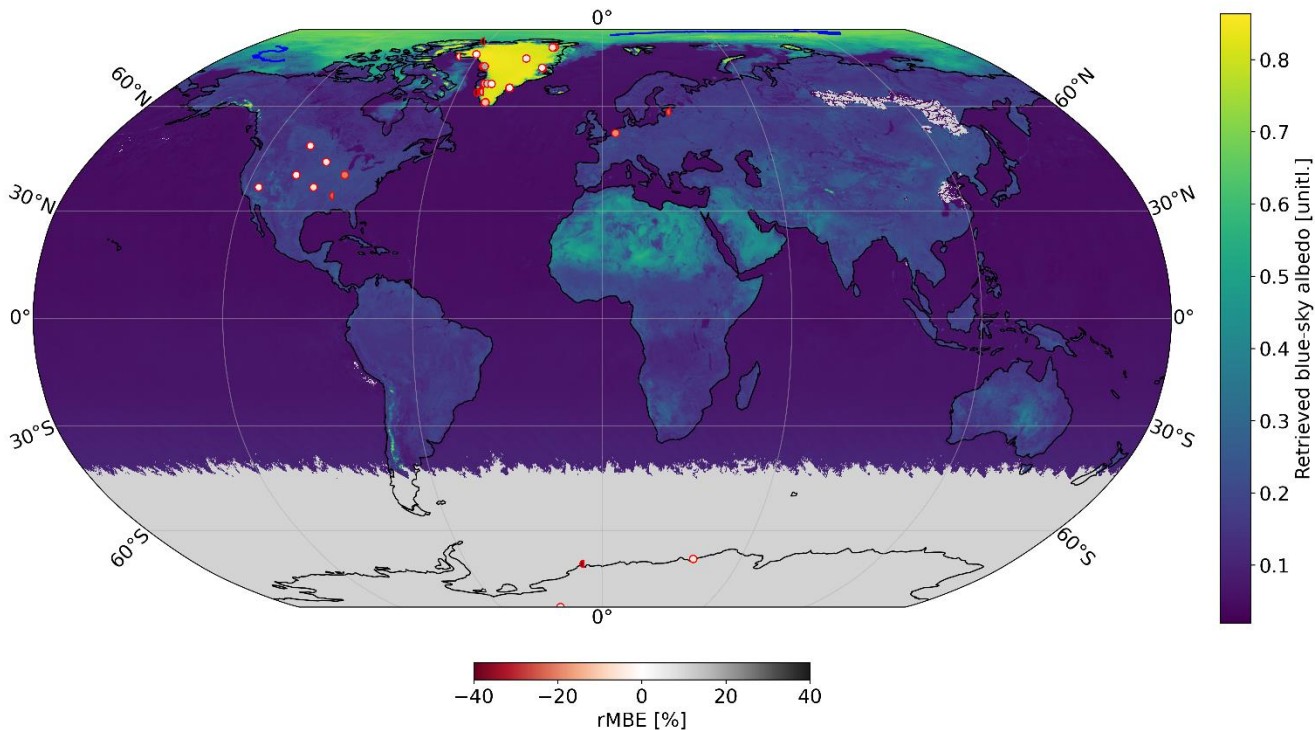

**Figure 7:** Example of CLARA-A3 blue-sky albedo (BAL), monthly mean of July 2018. Grey marks no-data regions. The overlaid circles indicate the overall relative bias (rMBE [%], bottom color bar) against reference in-situ observations from the BSRN and PROMICE station networks. Half-circles indicate reference sites that were evaluated but found spatially unrepresentative. Blue paths over the Arctic sea ice zone indicate drifts of the SHEBA and Tara ice camps

As a summary of the results, the mean relative bias was <10% over the multidecadal BSRN data coverage, with temporal trends in bias of less than 4% per decade (<2% per decade for several sites). This demonstrates that the AVHRR radiance intercalibration has achieved good stability during the BSRN coverage period. However, some emergent calibration issues have affected 2019-2020 and the subsequent ICDR (detailed in the Discussion section, Sect. 4). More variability across the validation sites was apparent in the assessed precision (bias-corrected RMS error as the metric) of the CDR. The principal cause is the coarse resolution of the AVHRR sensor and the resolution of the CLARA-A3 CDR (0.25° x 0.25° / 25 km x 25 km), which can easily cause issues in the spatial representativeness of in situ observations in a point-to-pixel validation of the surface albedo estimates (Róman et al., 2010; Riihelä et al., 2013). Significant effort was made to assess and quantify the representativeness of BSRN and PROMICE sites using Google Earth Engine's high-resolution Dynamic World land cover data (Brown et al., 2022). Fig. 7 shows the sites where analysis suggested poor representativeness as half-markers; full details are available in the VR and associated papers (accessible from the data record link given in Sect. 5).

## 3.4 Top of atmosphere radiation

The measurable quantities of the TOA radiation are the reflected solar flux, RSF, deduced from visible and near-infrared
AVHRR channels 1 and 2, and the outgoing longwave radiation, OLR, deduced from the infrared AVHRR channels 4 and 5
(see Table 1). Therefore, the total TOA radiation budget can be estimated from these quantities and the knowledge of the
incoming solar radiation at TOA (solar constant).

A full global coverage of broadband observations is provided by the Clouds and the Earth's Radiant Energy System (CERES)
instruments and derived products (Loeb et al., 2018), which are acknowledged to be the golden standard w.r.t. radiative flux
data records. However, there has been an increasing need for long-term, high-resolution TOA albedo products in monitoring
the climate impacts of regional-scale events such as air pollution, urbanization, forest fires, and other small-scale land cover
changes (Song et al., 2018), which can hardly be detected from data sets with coarse spatial resolution (Wang et al., 2016),
and small-scale atmospheric processes e.g. valley fog (Clerbaux et al., 2009). Furthermore, in absence of a global long-term
CERES-like CDR, many studies focusing on long term model validation or trend detection fall back to "surrogate datasets"
such as reanalysis (e.g. ERA-Interim) or radiative transfer computations (e.g. ISSCP), but would otherwise have preferred a
more observation-based alternative. Concerning CERES, two limitations can thus be identified: (1) the products are relatively
recent, e.g. starting in year 2000 for the EBAF product, and (2) the products have a relatively coarse spatial resolution of $1°x1°$
(Fig. 9a). The currently developed TOA flux products in CLARA-A3 resolve those two drawbacks, respectively by (1) a
prolongation back in time to the late 70's and (2) by increasing the spatial resolution to $0.25°x0.25°$ (Fig. 9b). A third advantage
of the new CDR's lies in their synergy and compatibility with the other CDR's from the CM SAF CLARA product family
(cloud mask and other cloud parameters, surface radiation, surface albedo, etc.) sharing common algorithms and processing
chains.

### 3.4.1 Methods and input data

The RSF retrieval is a 3-step process. Firstly, a spectral conversion is performed where narrowband reflectances from AVHRR
channels 1 and 2 are converted to broadband reflectance using empirical relations with the Clouds and the Earth's Radiant
Energy System (CERES) (Wielicki et al., 1996). The spectral conversion was accomplished using a large data set of collocated,
co-angular, and simultaneous AVHRR-CERES observations. It required knowledge of the specific orbital conditions for the
AVHRR- and CERES-carrying satellites. The analysis used all available data to achieve an unprecedented observation
matching between both instruments. Scene-type-dependent regression coefficients were obtained for a multivariate linear
regression model. The model development is partly based on existing literature and calibrated, validated, and documented by
Akkermans and Clerbaux (2020), where it is proven to be statistically robust and well-fitting. Besides the narrowband
reflectances, the model contains two more predictors which improve the regression model's accuracy: solar zenith angle and
viewing zenith angle, with the inverse of the cosine approximating the atmospheric optical path.


Secondly, an angular conversion is performed: anisotropy is corrected by applying scene-type-dependent Angular Distribution Models (ADMs), which convert broadband directional reflectance into hemispherical albedo (Loeb et al., 2003, 2005). The ADM scene type is a combination of land cover type and cloud properties (cover, phase, optical thickness), and in the case of clear-sky water, also wind speed. The scene type is selected using look-up tables in which surface types, wind speed, and cloud

parameters are classified in discretized bins, with each combination of bins leading to a different scene type. For every scene type, a three-dimensional data structure describes the expected reflectance and hemispherical albedo for all possible combinations of solar zenith angle, viewing zenith angle, and relative azimuth angle. Then, a trilinear interpolation between the angles is performed to estimate the resulting anisotropic correction factor required to convert the observed reflectance to hemispherical albedo. The resulting albedo is remapped from the GAC orbit grid to a nested 0.25°x0.25° lat-lon grid, in which

the grid boxes towards both poles (N and S) are systematically merged in the longitudinal direction to minimize areal distortions.

Thirdly, all instantaneous hemispherical albedos for a given day are temporally integrated, making it a multi-satellite product (if there are multiple satellite observations during that day). This is done using a diurnal cycle model, which considers the

(scene type dependent) relation between albedo and solar zenith angle. This method is called the "*constant meteorology method*", documented extensively by Young et al. (1998) and used subsequently in the CERES processing, where it is also called the "*CERES-only (CO) method*" (Doelling et al., 2013). The model produces a separate diurnal cycle associated with each instantaneous satellite observation (according to its scene type and scaled to fit the observation), then linearly interpolates between the two diurnal cycles associated with each pair of subsequent observations (Figure 8). As such, this flexible model

can ingest any number of observations at any time of day, making it suitable for any orbital configuration of NOAA and Metop satellites. The resulting interpolated diurnal cycle of albedo is converted to flux. This temporal interpolation is only performed for daylight conditions (when the solar zenith angle <84°). The flux during twilight conditions (when SZA>84°), prevailing near the terminator, is simulated with a separate, empirical model. The entire day is then integrated into a single daily mean RSF and subsequently in a monthly mean RSF. A detailed description of the entire RSF retrieval, including an overview of

the required input data, is published by Akkermans and Clerbaux (2021).

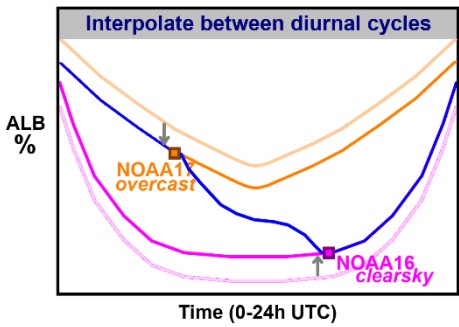

**Figure 8:** Conceptual illustration of the RSF temporal interpolation (figure not to scale). The average (expected) SZA-albedo curves associated with each observation's scene type are shown in light colors (orange and pink). These are scaled to match the respective observations, which is shown in dark colors (orange and magenta). The final interpolated diurnal cycle is shown in blue.

The OLR retrieval is a 2-step process. The first step is the estimation of the instantaneous OLR from the AVHRR observations in channels 4 and 5. This is done by regressions on the same large database of collocated AVHRR-CERES observations (as used for the RSF); for AVHRR/1 the regressions only make use of the channel 4 brightness temperature, for AVHRR/2 and AVHRR/3 both channel 4 and 5 are used (Clerbaux et al., 2020). In contrast to RSF, the OLR regressions are temporally varying (one for each month) and region-specific (one for each 5°x5° grid box), consist of an "all-in-one" conversion combining the spectral and angular corrections and use atmospheric reanalysis fields as additional predictors (humidity and temperature). Similar to the RSF retrieval, this is followed by a remapping from the GAC orbit grid to a nested 0.25° lat-lon grid. The second step concerns the estimation of daily and monthly OLR from the instantaneous AVHRR observations. Over clear sky land, the OLR from ERA5(T) reanalysis is used to estimate the diurnal variation; otherwise, simple linear regression is applied. A detailed description of the OLR retrieval, including an overview of the required input data, is published by Clerbaux et al. (2020).

### 3.4.2 Product examples and validation results

The level of detail of the CLARA-A3 product is demonstrated by zooming in on a particular region (Northern Atlantic and Europe) in Fig. 9, where the RSF product (bottom) is compared to the CERES (Wielicki et al., 1996) SYN1deg product (top).

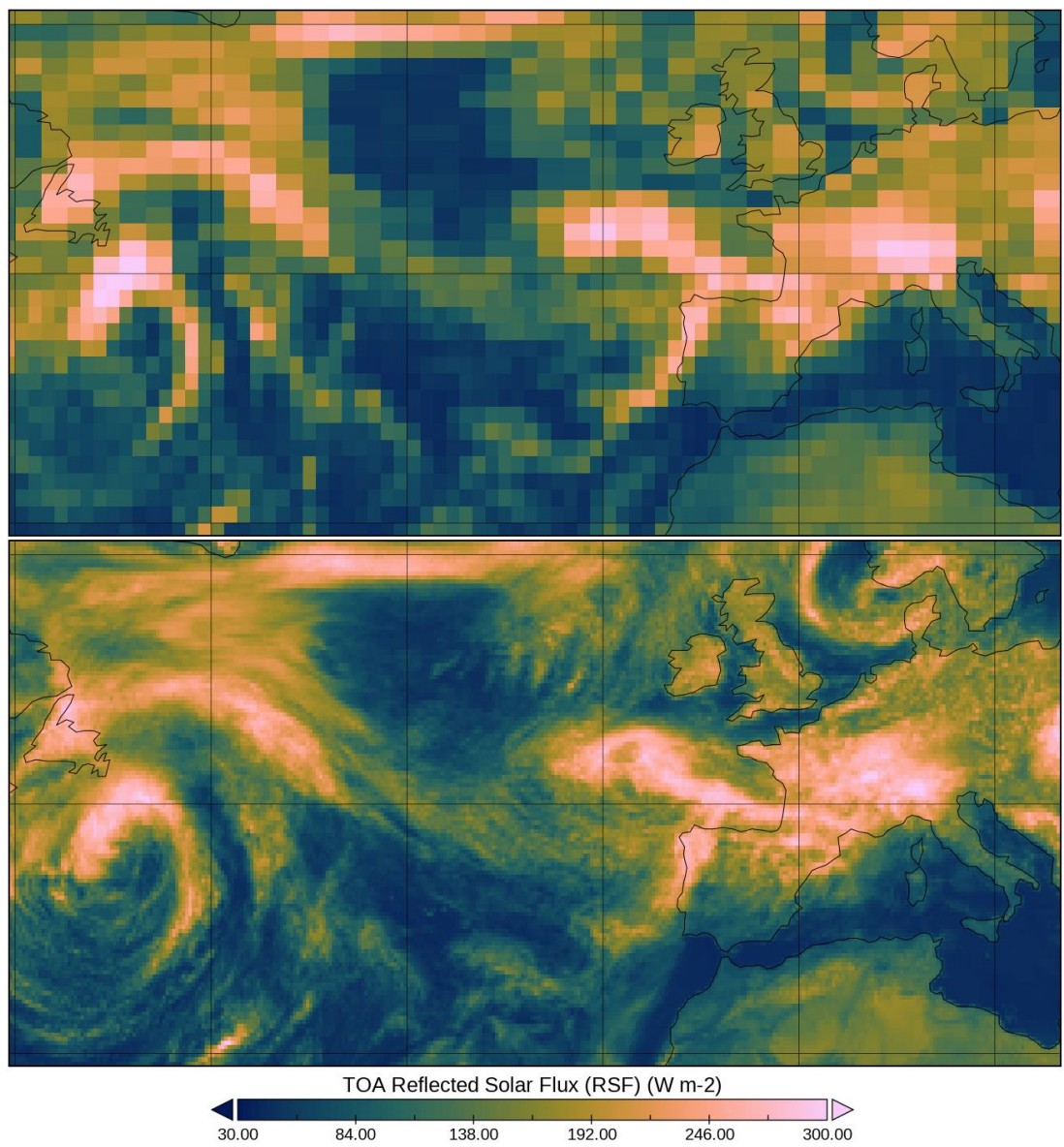

**Figure 9:** Daily mean RSF for 15-06-2008, zoomed in on the Northern Atlantic and Europe, from (a) CERES-SYN1deg, and (b) CLARA-A3

The intention here is not to provide details on the validation process or its results. The full description of the evaluation is available in the Validation Report (available from the link given in Sect. 5). What follows are some examples of the validation results. The global mean flux of monthly RSF from different data records is shown in Fig. 10, among which CLARA-A3 RSF is in orange.

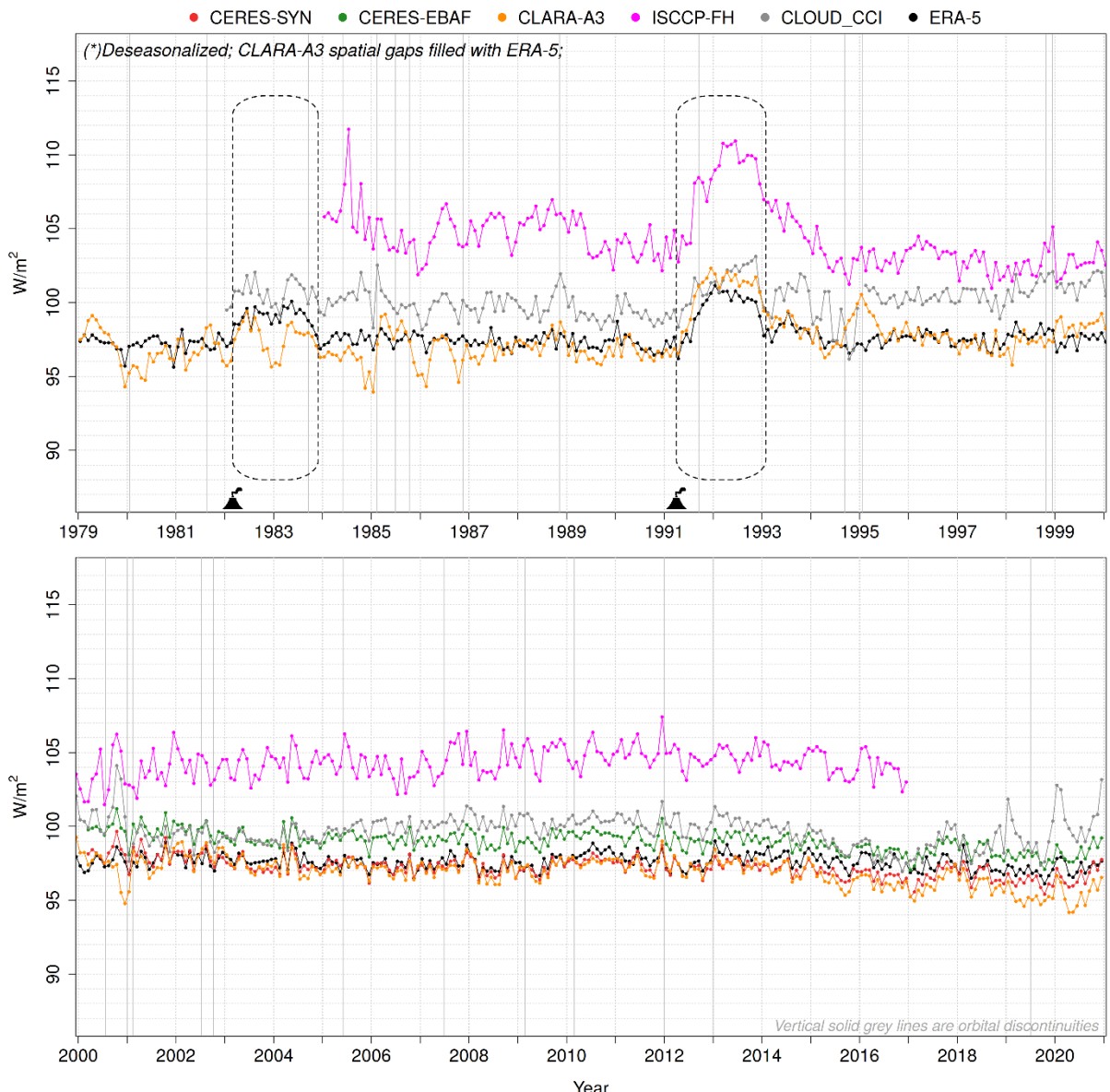

**Figure 10**: Global mean flux of monthly CLARA-A3 RSF and other data records. The periods with radiative impact from two major volcanic eruptions, El Chichón (1982/03-1983/12) and Pinatubo (1991/04-1993/01), are indicated.

The ERA5 time series (Hersbach et al., 2020) proves to be stable and can be used to assess the stability of other data records in the pre-CERES era (1979-1999). The two major volcanic eruptions, El Chichón and Pinatubo, are indicated on the time series, and their radiative impact is estimated at 3 and 5 W/m², respectively. The volcanic eruptions led to a dramatic increase in stratospheric sulphate aerosol loading, causing a considerable rise in the reflection of solar radiation due to the optical

properties of sulphuric acid droplets (Canty et al., 2013). Compared to CLARA-A3 and CERES-SYN, the RSF from CERES-EBAF is consistently about 1.5 Wm−2 higher (green curve in Figure 10), which can be explained by the EBAF adjustments made to comply with current consensus estimates of the global energy imbalance.

The CLARA-A3 Reflected Solar Flux data record is relatively stable as its bias w.r.t. ERA5 remains within a predefined envelope (mean +/- 2 W/m²) for 94% of the time. The largest biases of CLARA-A3 w.r.t. ERA5 are situated in the first decade of the data record. A suboptimal temporal coverage predominantly causes these since only morning or only afternoon satellites are available (see Fig. 1). During the CERES era (2000-2020), the CLARA-A3 RSF performance is very good, with a mean bias w.r.t. CERES-SYN1deg (red curve) close to zero for the larger part of the two decades, indicating good stability of the data record. Similarly, the global monthly mean OLR is also proven to be stable w.r.t. ERA5 as well as the observation-based HIRS-OLR data record (Lee et al., 2007, 2014), except for the first two years (TIROS-N and NOAA-6) (figures not shown here but details can be found in validation reports, see access link in Sect. 5).

Regional uncertainties are revealed by the Mean Absolute Bias (MAB) quantity and are closely related to the orbital configuration: better temporal coverage (i.e., more satellite overpasses per day) results in better performance (lower MAB). This is shown for the OLR in Fig. 11, where the best performance, with a monthly MAB around 1.5 W/m², is found during 2002-2016. Conversely, the MAB increased during periods with degraded temporal coverage, most notably the daily MAB with +40% for morning-only and afternoon-only orbits during 1979-1987. Furthermore, it is clear that the monthly MAB is systematically lower compared to the daily MAB.

The MAB for RSF shows similar patterns (figures not shown here but details can be found in validation reports, see access link in Sect. 5), with a monthly MAB of only 2 W/m² during 2002-2016, while the absence of a mid-morning orbit (before 2002) or early afternoon orbit (gradually after 2016) leads to a drop in performance (doubling of MAB). Table 4 summarizes the MAB results with reference to CERES and HIRS data over the entire CLARA-A3 period.

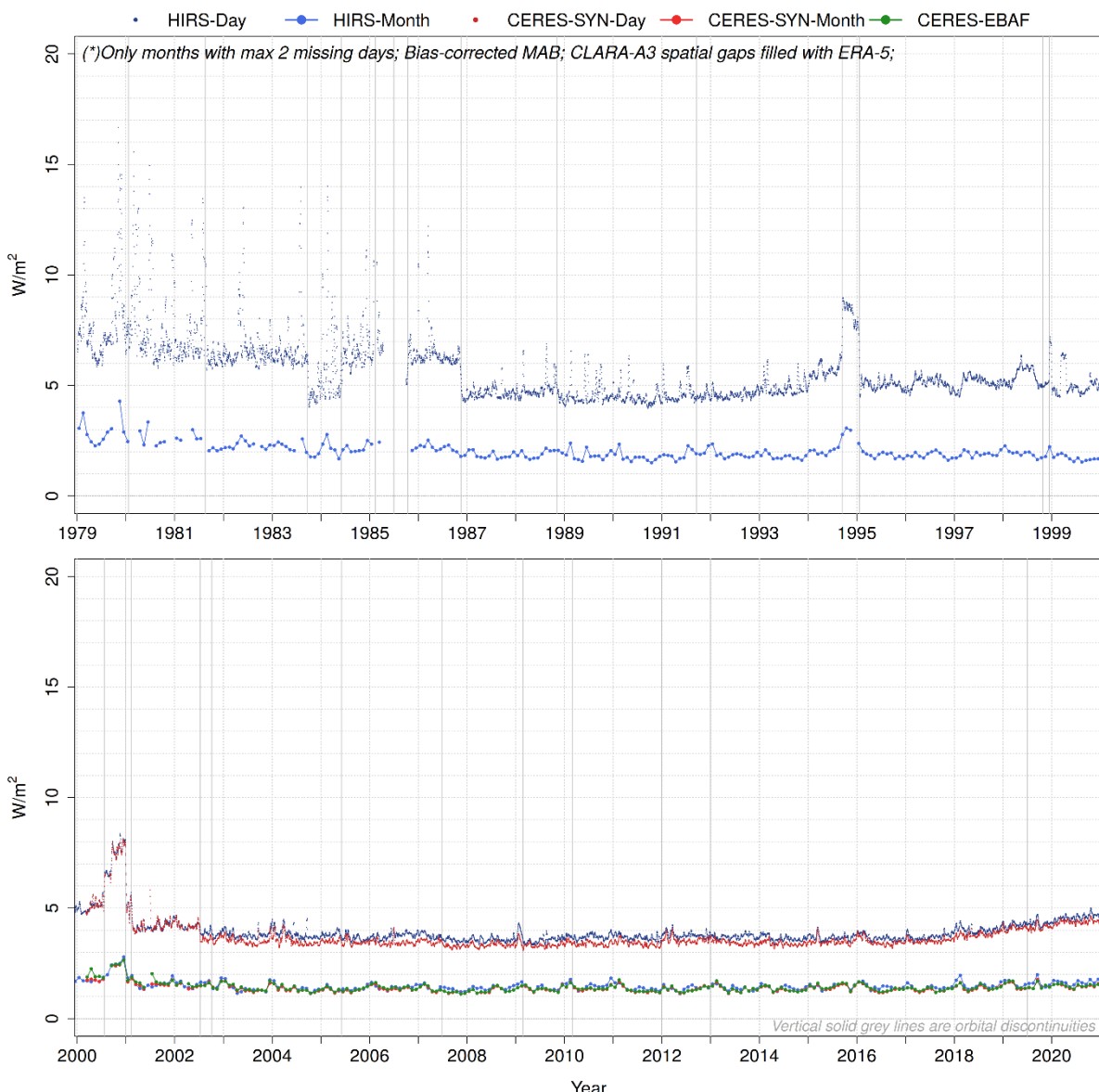

**Figure 11:** Global MAB between daily and monthly CLARA-A3 OLR and other data records. Daily MAB is systematically 2-3 $Wm-2$ higher compared to monthly MAB.

**Table 4:** MAB w.r.t. CERES and HIRS on both daily and monthly timescales, for both OLR and RSF.

|  | CLARA-A3 parameter | Time scale | MAB (W/m²) | Reference |
|---|---|---|---|---|
| CERES era (2000-2020) | Reflected Solar Flux (RSF) | Monthly | 2.3 | CERES |
|  |  | Daily | 6.2 |  |
|  | Outgoing Longwave Radiation (OLR) | Monthly | 1.4 |  |
|  |  | Daily | 3.7 |  |
| Entire record (1979-2020) | Outgoing Longwave Radiation (OLR) | Monthly | 1.8 | HIRS |
|  |  | Daily | 4.8 |  |

### 3.5 ICDR processing to extend the CLARA-A3 CDR

Previous sections have mentioned that ICDR generation has required some input data changes to enable processing. The main idea with the ICDR processing is to extend the CDR to allow assessment of the evolution of CLARA-A3 products beyond the end of the CDR until the present. For example, a user should be able to get the most recent results (e.g., for the latest month), which, for example, could be compared to the results of the CDR in anomaly plots. This could support the work with operational climate monitoring performed at national meteorological institutes.

However, producing the CLARA-A3 products in near-realtime is challenging. Given constraints on calibration quality and timely availability of input data, it cannot be done with exactly the same accuracy of products as for the CDR. Thus, ICDR products are truly interim products that eventually have to be replaced by high-quality products from the next edition of the CDR. This is best understood from the fact that the calibration of visible AVHRR channels is based on vicarious calibration methods requiring access to several years of data to accurately estimate calibration parameters (as described in Sect. 2). Consequently, particular problems could arise for 'young' satellites, i.e., satellites with a short measurement record where correction factors for temporal degradation are uncertain. In this case, the Metop-C satellite (launched in 2018) has shown such uncertainties in its calibration that it was decided to start ICDR production without this satellite. Metop-C will be re-entered into the ICDR once more reliable calibration information becomes available.

Another problem for ICDR production is that some ancillary datasets, used as input to retrieval algorithms for the CDR, cannot be accessed with short latency. Consequently, the ICDR production uses the preliminary ERA5T reanalysis dataset instead of ERA5. For similar reasons, information on ice cover is slightly different from the CDR. Therefore, extensive inter-comparisons of CDR and ICDR results for overlapping periods of the last year of the CDR (2020) were performed to assess the impact on

products. The differences were generally small and mostly confined to small geographic regions (e.g., marginal ice zone) or caused primarily by the exclusion of Metop-C in generating the ICDR level-3 products by reducing the temporal and spatial coverage. More information about the differences in input data for the CDR and ICDR can be found in the PUM document, and results from the CDR-ICDR inter-comparisons are described in the VAL document. Both documents are available via the data record link given in Sect. 5.

## 4 Discussion

In the following, we discuss some further aspects of the CLARA-A3 CDR, both regarding the impact of improvements and remaining problems.

Preferably, a CDR should be compiled from climate-quality, inter-calibrated and homogeneous radiances, formally denoted Fundamental Climate Data Records (FCDR). However, a complete AVHRR GAC FCDR as a basis for retrieval of various ECVs is still missing, but actions have been taken to take care of known issues with the basic radiances. The compilation of the EUMETSAT AVHRR FDR has enhanced the quality control procedures compared to previous CLARA editions, and now the FDR will also be made available to users other than the CLARA team. However, one problem with the AVHRR information from the visible channels is that accurate calibration can only be achieved retrospectively by applying vicarious calibration techniques.

The FDR uses the 2017 calibration update of the visible calibration, which means that calibration information needs extrapolation after 2017, for 2018-2020 in the CDR, and for the first years of the ICDR. This is problematic for satellites with a relatively short period of operations (e.g., Metop-B), where the time-dependent calibration corrections will be less reliable than those with longer operation periods. For the CDR, a special solution had to be used for the Metop-C satellite launched in 2018. A preliminary calibration correction for this satellite has been made available by NOAA to include Metop-C data for 2019 and 2020. However, analysis proved that the time-dependent corrections of Metop-C were very uncertain, and, therefore, the ICDR production in 2021 started without this satellite A new calibration update from NOAA is expected in the near future. Thus, improved calibration for the ICDR will be possible after the availability of this update. At this point, we foresee re-introducing Metop-C among the used satellites. In the longer term, the number of satellites with the original AVHRR/3 instrument will gradually decrease and eventually disappear. The aim is to replace original AVHRR-data in future CLARA editions with data from AVHRR-heritage channels from the Visible Infrared Imaging Radiometer Suite (VIIRS) sensor on the NOAA Joint Polar Satellite System (JPSS) satellites and from the METimage sensor on the EUMETSAT Polar System – Second Generation (EPS-SG) satellites. The first upgrade of the CLARA dataset in this respect is planned for 2026 and will use data from VIIRS.

As seen in Sects 3.1.1 and 3.1.2, the overall CLARA-A3 capacity to characterize the global 3-dimensional distribution of clouds (i.e., here restricted to the 2-dimensional distribution of cloudiness and the uppermost cloud top levels) and its evolution over the 42 years has considerably improved compared to the predecessor CLARA-A2. A major improvement is in the cloud top height estimation, which is now much more realistic and is actually the closest among all evaluated data records to the CALIPSO Top Layer data record, which is assumed to provide the most accurate description of global cloudiness. The results over all four decades and compared to the other existing data records show that the CLARA-A3 data record currently gives the best estimate of global 3-dimensional cloud distributions. PATMOS-x results are close, but for CTP, there is no sign of any seasonal variation of cloud tops, which is very significant for both CLARA-A3 and MODIS data records. Some seasonal variation (with the highest cloud tops in the Northern Hemisphere summer) should be expected when considering the different cloudiness behavior of the summer monsoons linked with the oscillating ITCZ in the two hemispheres. The amount and intensity of convective clouds associated with the summer monsoon is clearly larger in the northern hemisphere which is partly explained by more land masses being affected by the monsoon (e.g., India and Southeast Asia). It is worth mentioning that CLARA-A3 lacks information on multi-layer cloudiness in contrast to some other data records. On the other hand, such information can only cover the special case of very thin cirrus clouds overlying thicker water clouds. All other multi-layer cloud situations are impossible to observe from AVHRR data. A deeper discussion on cloud top distributions and their seasonal variability can be found in Karlsson et al. (2023a).

Despite the cloud dataset improvements, new and problematic issues have emerged for CLARA-A3 not seen before. For the first time, data from the earliest AVHRR/1 version of the instrument have also been included in the data record. Whereas cloud detection appears to work relatively well for this version (although with some tendency of cloudiness overestimation in Fig. 2), cloud top estimation appears to be very different (i.e., with higher cloud top altitudes) for satellites TIROS-N and NOAA-6 compared to results for the other two versions of the AVHRR instrument (Fig. 3). Some differences could be expected since methods can only use two of three possible infrared channels. Consequently, for cloud top height retrievals, the 11 µm channel is used together with the 12 µm channel for AVHRR/2 and AVHRR/3, while for AVHRR/1, the 12 µm channel is replaced with the 3.7 µm channel. However, after processing simulated AVHRR/1 data based on AVHRR/3 data (by excluding the channel missing from AVHRR/1 from AVHRR/3), only marginal differences in retrieved cloud top results could be seen when comparing with results from the full five-channel instrument (for full details, see VAL report with the link given in Sect. 5). Thus, the deviation is likely to come from the data itself and not primarily from the method used. Indeed, further analysis of the AVHRR FDR dataset indicated suspiciously low infrared brightness temperatures for the TIROS-N AVHRR when compared with corresponding measurements from the HIRS 11 µm channel (EUMETSAT, 2023). Thus, we conclude that further work is needed to understand better and possibly improve the infrared calibration of data from AVHRR/1.

The retrieval of 'downstream' cloud products has benefited from improved cloud masking and cloud top height determination compared with CLARA-A2. In particular, the fraction of ice clouds has increased globally and become closer to CALIPSO

observations, following increases in high cloud fraction. However, some fundamental limitations of the cloud optical and microphysical property products have remained. These include sensitivity to the satellites' orbital drift and to the variable use of channels 3a and 3b during daytime. An attempt has been made to mitigate the latter effect by only including optical and microphysical properties from AVHRRs with channel 3b active in the merged level-3 products, leading to improved stability

of, in particular, effective radius (see Figure 5-36 in the CLARA-A3 Cloud Products validation report, available through the link given in Sect. 5).

The provision of white- and blue-sky albedo estimates now allow for accounting of atmospheric effects in surface albedo for surface energy budget studies involving CLARA data. Also, land surface models used in, e.g., climate models, often

parameterize black- and white-sky albedos separately, against which the CLARA-A3 should now be comparable (Oleson et al., 2003). Surface albedo data in past CLARA editions have seen notable uptake in cryospheric studies (e.g., Karlsson and Svensson, 2013, Cao et al., 2015, Thackeray and Hall, 2019); validation results from CLARA-A3 indicate equivalent or improved performance over snow and ice, suggesting that the new extended edition could continue to serve in this role in, e.g., mapping of albedo trends over terrestrial and oceanic cryospheres, and in calculations of surface albedo feedback.

For TOA RSF and OLR, periods with higher uncertainty (increased MAB) can be explained by a degrading temporal coverage which introduces biases in regional climates with an asymmetric diurnal cycle (e.g., marine stratus thinning or land convection), depending on the region, season, and kind of phenomena. Furthermore, a degrading temporal coverage also introduces biases with fast moving small-scale or heterogeneous weather systems (e.g., fronts), typically consisting of swirls

with positive alongside negative bias, caused, for example, by an extrapolation of the morning observation to the afternoon (without observation, typically occurring in the period 2012-2020 due to orbital drift of satellites NOAA-18 and NOAA-19). The daily MAB is generally higher than the monthly MAB because some biases vary in sign daily. Consequently, they tend to compensate each other on a monthly time scale. This is the case for fast-moving, small-scale, or heterogeneous weather systems. It also occurs for some biases in the instantaneous flux retrievals due to errors in ADM related to viewing and

illumination geometry and scene type identification, such as cloud cover and cloud properties.

Overall, the CLARA-A3 TOA RSF and OLR validation results are satisfactory for their first edition. Given the suboptimal orbital configuration of only a morning or an afternoon satellite during the first years of the record (1979-1987), caution is advised when including this period for climate trend analyses (although it is still useful for other purposes). Furthermore,

uncertainties inherent to the polar orbiting satellite constellation are difficult to correct, especially for a constellation with persisting orbital drift, as is the case with most NOAA satellites; this is in contrast to the constant local observation time (equatorial overpass time) of the Aqua and Terra satellites, which allowed the development and implementation of a fixed instantaneous-to-diurnal correction for the CERES products. However, some potential improvements for future CLARA editions can be noted:

1. Updating the currently implemented CERES Ed2 ADMs to the newest available CERES Ed4 ADMs could improve the instantaneous RSF estimation, as well as the albedo diurnal cycle models.

2. The orbital drift effects of the last afternoon orbit (NOAA-19) could be solved by introducing the VIIRS instrument (in afternoon orbit) alongside the existing AVHRR-carrying orbits.

3. An update of the AVHRR FDR with the newest calibration could solve calibration issues with the most recent satellites, such as Metop-B and Metop-C.

4. Corrections to the flux data records could be added to close the energy budget and comply with existing consensus estimates on the energy balance or potential imbalance.

## 5 Data

The data record doi for CLARA-A3 is 10.5676/EUM_SAF_CM/CLARA_AVHRR/V003 (Karlsson et al., 2023b). Data and associated documentation (scientific references, algorithm theoretical basis documents, validation reports and user manuals) are available through the following link:

https://doi.org/10.5676/EUM_SAF_CM/CLARA_AVHRR/V003.

The AVHRR FDR data record used as a basis for compiling the CLARA-A3 CDR can be accessed with the data record doi 10.15770/EUM_SEC_CLM_0060.

All intellectual property rights of the CM SAF CLARA-A3 products belong to EUMETSAT. The use of these products is granted to every interested user, free of charge. If you wish to use these products, EUMETSAT's copyright credit must be shown by displaying the words "copyright (year) EUMETSAT" on each of the products used

## 6 Conclusions

From extensive validation efforts of the CLARA-A3 CDR, we demonstrated that, although not tailored for climate monitoring, the AVHRR sensor provides precious information about several essential climate variables (ECVs) and their temporal evolution. Especially, the third CLARA edition has taken further steps to improve the climate data record by

1. Extending the temporal coverage both backward in time (to 1979) and forward in time (to 2020),

2. Introducing an extension of the dataset after 2020 by a continuous production of a CLARA-A3 ICDR,

3. Enhancing the quality control of AVHRR GAC radiances to identify and exclude problematic parts of the dataset,

4. Updating most of the involved retrieval methods, both in terms of giving more accurate results but also by providing better uncertainty estimates,

5. Adding additional components to the surface radiation products enabling the estimation of the total radiation budget at the surface,

6. Adding two more flavors of the surface albedo product,

7. Adding TOA radiation budget products securing that the CLARA data record can give a complete picture of atmospheric and surface radiation conditions.

The upgrade of the CLARA data record marks a significant enhancement that enables a more complete description of cloudiness, surface albedo and radiation conditions at the surface and in the atmosphere. The access to cloud parameters and surface albedo estimations parallel to the radiation products makes the CLARA-A3 data record suitable for further studies on potential feedback effects in the radiation climate caused by ongoing climate change. The new CLARA-A3 edition has also led to updates of the COSP simulator for the CLARA data record (Eliasson et al., 2020) for use in climate model inter-comparisons.

CLARA-A3 encompasses more than four decades (1979-2020) of global observations, and a system for continuous upgrades of ICDR versions of the data record ensures a regular extension. ICDR products will be continuously monitored with relevant reference datasets where also new datasets like GCOM-C (Global Climate Observation Misson - Climate, Nakajima et al., 2019) and CARE (The Cloud remote sensing, Atmosphere radiation and Renewal Energy application product, Ri et al., 2022) will be considered. Further extensions of CLARA into the future will be possible even if the data record with AVHRR data will end with the Metop-C satellite (currently operating since 2019). AVHRR-heritage channels on the modern VIIRS and METimage sensors can be utilized to further extend the data record for another 2-3 decades.

**Author contributions**

KGK prepared the original manuscript with substantial contributions from JFM, AR, JT and TA. KGK, AD, SE, EJ and NH developed and validated the methods for cloud detection and cloud top height retrievals. JFM and NB developed and validated the methods for the cloud microphysical products. AR developed and validated surface albedo products. JT developed and validated surface radiation products. TA and NC developed and validated the TOA radiation products. MSt, IS, AR, JT and TA developed and implemented methods for level-3 calculations. DS and NS were responsible for computer operations and the final compilation of products. MSc and RH provided valuable comments and recommendations for the structure of the manuscript. All authors contributed to the manuscript.

**Competing interests**

The authors declare that they have no conflict of interests.

**Acknowledgements**

We are grateful to Dr. Andrew Heidinger at NOAA/NESDIS/STAR, Madison, Wisconsin, and Dr. Mike Foster at Cooperative Institute of Meteorological Satellite Studies, University of Wisconsin–Madison, Madison, Wisconsin, for support concerning the calibration of AVHRR GAC data.

We also acknowledge Marie Doutriaux Boucher and her expert team at the EUMETSAT Secretariat for compiling the AVHRR FDR data record.

The CALIOP V4.20 data were obtained from the NASA Langley Research Center Atmospheric Science Data Center at https://asdc.larc.nasa.gov/project/CALIPSO.

**Financial support**

This work was performed within the EUMETSAT CM SAF framework and all authors acknowledge the financial support of the EUMETSAT member states.

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
