# Peer review of "CLARA-A3: The third edition of the AVHRR-based CM SAF climate data record on clouds, radiation and surface albedo covering the period 1979 to 2023"

_Earth System Science Data, 2023_

## Author Comment (AC1)

Reply to Referee #1 on the Earth System Science Datasets manuscript

**” CLARA-A3: The third edition of the AVHRR-based CM SAF climate data record on clouds, radiation and surface albedo covering the period 1979 to 2023 ”**

**by**
**Karlsson et al, 2023**

**Repeating general comment:**

**This paper describes the third edition of the product (CLARA-A3), including surface albedo, surface radiation, and TOA radiation budget products ranging from 1979 to 2020. Various validations for cloud and radiation products were done, and show a good agreement with other or ground-based measurements. Overall, this manuscript is clear. However, there are several issues that need to be taken care of before this paper becomes acceptable for publication.**

**Reply:**

Answers to the referee's comments and questions are given below together with suggestions on how to improve and update the manuscript.

**Detailed comments:**

1. L285, Figure 2. The cloud fraction from CLARA-A3 and previous version CLARA-A2 show a large difference, especially overestimation of cloud fraction, why?

   **Author reply:** The reviewer's interpretation is not correct. Cloud amounts in CLARA-A3 are improved (and not overestimated) in comparison to those in CLARA-A2. This is also clearly stated on lines 272-278. More details on the validation results can be found in the CLARA-A3 Validation Report and in a recent publication by Karlsson et al., 2023, in the Remote Sensing journal (https://www.mdpi.com/2072-4292/15/12/3044).

   Cloud amounts in CLARA-A2 suffered in particular from underestimated cloud amounts over the polar regions during the polar winter. This is described by Karlsson and Håkansson, 2018 (https://amt.copernicus.org/articles/11/633/2018/).

2. How about the cloud fractions over polar regions? I am curious about these products' performance over polar regions between snow/ice and cloud detections.

   **Author reply:** As mentioned in the previous point, previous cloud amounts over the polar regions in CLARA-A2 suffered from large underestimations in the polar winter (https://amt.copernicus.org/articles/11/633/2018/). However, after repeating the same validation effort against CALIPSO-CALIOP data for CLARA-A3, it is clear that polar winter results have improved for the Arctic region (see in particular Figure 4 in https://www.mdpi.com/2072-4292/15/12/3044). However, results over Antarctica have not improved in the same manner and it is clear that conditions are more challenging here. The problems over extremely cold surfaces at night are well-known (and encountered for most passive imagers) and it is clear that a sensor like AVHRR does not provide enough of useful information for being able to perform an efficient cloud screening under such conditions. Daytime results are, however, much more reliable indicating that the snow-cloud discrimination problem is manageable for solar illuminated surfaces. This is important for e.g. the estimation of the surface albedo product in the polar areas.

   We suggest to add a few sentences about this and more clearly refer to the paper in Remote Sensing for more details.

3. Why are there many invalid values of surface albedo in Eastern China?

**Author reply:** The atmospheric correction necessary for the surface albedo retrieval becomes uncertain over areas with high aerosol loading in the atmosphere. In practice, we discard all land surface observations with AOD > 1.0 at 550 nm. This limit is generally exceeded over both Eastern Siberia and Eastern China over much of the summer period (see attached figure for the monthly mean AOD for July as used in the retrievals for 2016 as an example). Therefore, most of the observations are discarded, although some data may still end up being accepted if it originates over large rivers and lakes and is therefore identified as water albedo, which follows a separate model-based retrieval path where AOD restrictions do not apply.

[Figure]

There is content on lines 513-515 about the influence of AOD, we could simply amend the sentence there to say "(Siberia and Eastern China in the example)".

4. L610, Figure 10. For global mean flux, the CLARA-A3 RSF is close to CERES-SYN, both underestimation of CERES-EBAF, why? And the ISCCP-FH show totally overestimation.

**Author reply:** The referee identifies some remarkable differences between some of the data records, and we agree that some of it may be briefly explained in the text. The CERES-EBAF product differs from the CERES-SYN in the way the diurnal cycle is derived (it makes use of so-called diurnal asymmetry ratio's instead of simple matching to the observed diurnal cycle from geostationary satellite (GEO), avoiding the typical GEO edge artefacts) but more importantly for your question, CERES-EBAF is also subject to some a posteriori "tuning" to match the scientific consensus regarding global energy imbalance measurements (as derived from ocean heat data records). Mainly the latter is responsible for the offset between the two products that we notice in Figure 10. Furthermore, indeed, ISCCP-FH shows a large overestimation: it should be stressed that for ISCCP-FH the fluxes are not directly observed but calculated by radiative transfer models and using cloud observations as main input (similar like the CLOUD_CCI approach). But for this, the choice of radiative model as well as the quality of the input data is important for the end result, and also the treatment of processes like diurnal cycle interpolation and orbital drift corrections. As far as we know this is not taken into account in ISCCP-FH. However, we don't feel we should elaborate on that in the text. So, to conclude, as a response to the author's comment, we suggest adding the following sentence in L610:

"Compared to CLARA-A3 and CERES-SYN, the RSF from CERES-EBAF is consistently about 1.5 Wm$-2$ higher (green curve in Figure 10), which can be explained by the EBAF adjustments made to comply with current consensus estimates of the global energy imbalance."

5. The algorithm of radiation updated from CLARA-A2 to CLARA-A3 is not clear, authors should present more details in this paper.

**Author reply:**
The basic algorithm to estimate surface solar radiation from the AVHRR data has not been updated from CLARA-A2 to CLARA-A3. The changes and improvements in the data quality of the CLARA-A3 surface solar radiation data record (compared to CLARA-A2) can be attributed to the use of improved input data, i.e., the probabilistic cloud mask (compared to the binary cloud mask used in CLARA-A2) to distinguish between clear

sky and cloudy pixels, and the reflected solar radiation flux, which had been estimated with a very basic method in CLARA-A2 and is now being derived with a much more advanced algorithm (see the Section on the top-of-the-atmosphere radiation). In addition, the use of surface albedo from ERA-5 (compared to a climatology used in CLARA-A2) also improves the surface radiation estimation for cloudy and clear-sky pixels.

These changes will be stated more clearly in the revised version of the manuscript.

6. The algorithm of instantaneous radiation converting to daily mean is very important (Eq. 1), did the authors try another interpolation algorithm? Such sinusoidal fit, or Wang & Pinker method?

**Author reply:**
We agree that the calculation of daily averages from the instantaneous retrieval of surface solar radiation is critical for the accuracy of the daily (and subsequently the monthly) surface solar radiation data record. As described our method uses the daily mean of the clear-sky surface radiation (derived with a clear-sky solar radiative transfer model) as a 'first estimate' of the all-sky surface solar radiation. By weighting the clear-sky daily mean with the ratio of the sum of all all-sky observations to the sum of the corresponding clear-sky estimations this method easily handles a variable number of observations and provides good estimates also for very few observations (In the case of a clear-sky day, even one single observation is sufficient!).

The method by Wang and Pinker, 2009, has been designed for MODIS observations with two observations at given times; its applicability to the AVHRR instruments, with changing number of available instruments over time appears complex. In addition, the Wang and Pinker approach, as well as the sinusoidal fit, only considers the incident radiation at the top-of-the-atmosphere (represented by the solar zenith angle) for the scaling of the radiation estimates. In contrast, the use of the daily mean clear sky surface solar radiation (as in the approach used in the CLARA data records) is better representing the spatial (e.g., varying surface albedo) and the temporal (e.g., varying water vapor) variability on the daily average surface solar radiation, which is expected to provide more accurate estimates even though no detailed comparison between different methods to derive daily averages from polar-orbiting satellite instruments have been conducted so far.

7. How did the authors get daily mean longwave radiation at the surface from instantaneous data?

**Author reply:**
Daily mean surface longwave radiation is not derived within the CLARA retrieval algorithm; this will be stated more clearly in the revised version of the manuscript. The surface longwave radiation in the CLARA-A3 data record is only estimated and provided as monthly averages. These data are closely linked to the corresponding estimates from ERA-5.

8. It is recommended to introduce the current status and characteristics of relevant cloud and radiation products in the introduction of the paper, and highlight the advantages and characteristics of the new product (such as long time series characteristics and fine cloud and radiation parameters). Related products including JAXA's GCOM-C product (https://suzaku.eorc.jaxa.jp/GCOM_C/) (Nakajima et al., 2019), CARE (http://www.slrss.cn/care/) cloud characteristics and radiation products (Letu et al., 2020, 2022; Xu et al., 2022). This product is based on the latest geostationary satellites spliced together to form a high spatiotemporal resolution and high-precision remote sensing product.

**Author reply:** Our opinion is that we have introduced the CDR in a reasonable way, emphasizing the important changes that were introduced with references to previous editions (lines 65-77). Furthermore, for every sub-section related to individual product components, improvements and changes have been described with references to earlier works and validation studies. We feel that a major revision of the introduction section, in order to give a richer background and status description, will automatically also lead to the need for a major revision of all individual sub-sections. We ask the reviewer to reconsider this recommendation based on the described consequences which could substantially delay the final publication.

Regarding the recommendation to also compare products to the mentioned additional datasets GCOM-C and CARE, we have to say that the core CLARA-A3 dataset (i.e., not including ICDR data) ends in 2020 and our main task has been to use available global long-term datasets on (at least) the decadal scale to validate our products. The two mentioned datasets have either not covered long enough time to be considered (CGOM-C) or they have not provided global coverage (CARE). Nevertheless, the two datasets are certainly interesting for the near future

regarding the continuous monitoring of the ICDR products. We propose to add the following statement (e.g., after the first sentence in the paragraph starting on line 805):

"ICDR products will be continuously monitored with relevant reference datasets where also new datasets like GCOM-C (Global Climate Observation Misson - Climate, Nakajima et al., 2019) and CARE (The Cloud remote sensing, Atmosphere radiation and Renewal Energy application product, Ri et al., 2022) will be considered."

Suggested additional references:

Nakajima, T. Y., Ishida, H., Nagao, T. M., Hori, M., Letu, H., Higuchi, R., ... & Yamazaki, A. (2019). Theoretical basis of the algorithms and early phase results of the GCOM-C (Shikisai) SGLI cloud products. Progress in Earth and Planetary Science, 6(1), 1-25.

Ri, X., Tana, G., Shi, C., Nakajima, T. Y., Shi, J., Zhao, J., ... & Letu, H. (2022). Cloud, Atmospheric Radiation and Renewal Energy Application (CARE) Version 1.0 Cloud Top Property Product From Himawari-8/AHI: Algorithm Development and Preliminary Validation. IEEE Transactions on Geoscience and Remote Sensing, 60, 1-11.

---

## Author Comment (AC2)

Reply to Referee #2 on the Earth System Science Datasets manuscript

**” CLARA-A3: The third edition of the AVHRR-based CM SAF climate data record on clouds, radiation and surface albedo covering the period 1979 to 2023”**

**by**
**Karlsson et al, 2023**

**Repeating general comment:**

**The authors describe an updated AVHRR-based cloud, radiation, and albedo record CLARA-A3. The AVHRR instruments have made observations from low Earth orbit since the late 1970s and provide a valuable cloud-climate record for the scientific community. However, there are many problems with this instrument: poor calibration, changing instrument characteristics over time (SRFs, channel choices, etc.), challenges with orbital drift, and sparse coverage of the solar and thermal plank functions, to name a few. The authors have been working on this observational record for a long time and should be commended for getting the most out of it. CLARA-A3 has several additional variables compared to the previous version, and the variables common across earlier CLARA versions have shown improvement with respect to other passive and active satellite records. My comments and suggestions regarding the manuscript are fairly minor but should be considered in a revision by the authors.**

**Reply:**

The authors are grateful to the referee for these encouraging words. We have addressed comments and suggestions below and proposed relevant changes to the manuscript where needed.

**Detailed comments:**

1. Throughout the manuscript: At times the use of English can be fairly colloquial and could be tightened up.

   **Author reply:** OK, we will try revising the text where we find such parts (e.g., partly guided by some or your specific recommendations below).

2. L44: extra parenthesis **Author reply:** OK, will be deleted.

3. L46: probably should distinguish between individual VIIRS and MODIS records, and the VIIRS+MODIS continuity product, there should be more recent references for the latter

   **Author reply:** Good point. We propose adding the following at line 46:

   "In addition, measurements from the Visible Infrared Imaging Radiometer Suite (VIIRS) instrument is now capable of providing decadal scale observations suitable for CDR generation and they will also be used to extend the MODIS CDR through a specific VIIRS+MODIS continuity product (Platnick et al., 2021)."

   Added reference:

   Platnick, S.; Meyer, K.; Wind, G.; Holz, R.E.; Amarasinghe, N.; Hubanks, P.A.; Marchant, B.; Dutcher, S.; Veglio, P. The NASA MODIS-VIIRS Continuity Cloud Optical Properties Products. Remote Sens. 2021, 13, 2. https://dx.doi.org/10.3390/rs13010002

4. L79-83: The authors do point out that the reader could go to a validation report for more information. But it would be helpful to explicitly describe the comparisons against CERES fluxes. Since CERES has broadband channels, there are very few photons missed, whereas AVHRR has discrete SRFs and does not measure all frequencies. Also, AVHRR shows really nice spatial structure. It would be worth detailing why one would want to do this with AVHRR data in the first place.

   **Author reply:** In the lines L79-83 there is no reference to the validation report, and we are not sure which of the CLARA-A3 products the reviewer is talking about (surface fluxes? TOA fluxes?). For the TOA fluxes, the comparison against CERES fluxes is performed and described

(Figures 9, 10, 11 + accompanying text). About added value of this data record: see further, remark on Figure 10.

5. L91: period at start of sentence **Author reply:** Will be deleted.

6. L91: 'has been measuring' should be e.g. 'has been making measurements'

   **Author reply:** OK, will be changed.

7. L131: period and comma next to each other **Author reply:** Period to be deleted.

8. L160: it should be made clearer that CLARA-A3 is Level 3, but what is Level 2b? Or is CLARA-A3 also Level 2b? This needs clarification.

   **Author reply:** In fact, for some products (i.e., exclusively cloud property products) also Level 2b representations are provided in addition to L3 products. This is clearly stated on lines 158-160 and in the following lines 161-163 the Level 2b concept is introduced and explained. We think it would be enough to just clarify that these products are not only useful and needed for L3 calculations but they are also available to external users. Consequently, the text "All level-2b products are also available for external users." is added at line 164 (after "products.").

   The motivation for making also Level 2b products available for users has to do with the fact that the CLARA-A3 simulator package (mentioned on lines 197-198) produces results from NWP/Climate models in the form of Level 2b products. Thus, results simulated from models can be compared both to instantaneous observations (Level 2b) and to aggregated results (Level 3).

9. L173-174: prepared separately from the rest of the globe? It would be better to simply say "This latitude band is prepared with gridding strategy A, while that latitude band is prepared with another gridding strategy B." or similar.

   **Author reply:** Well, we would prefer the following slightly changed formulation:

   "CFC and CTO products are prepared for two additional areas which cover and zoom in on the polar regions. This is motivated since the standard latitude-longitude grid is not appropriate for studies focused on the polar regions because of the variable geometric grid resolution near

the poles in the standard grid. The two polar regions (named South Pole and North Pole) have constant 25 km grid resolution and are used exclusively for the mentioned cloud products and for the surface albedo products (discussed in Sect. 3.3)."

10. L175: why does the sentence end with 'here'? **Author reply:** Deleted because of the reformulation proposed in the previous point.

11. L188: also, clause after comma starts with 'here', not sure what is meant

**Author reply:** "Here" refers to the definition of COT in CLARA-A3, which is at a wavelength of 0.64 μm. However, this information is not really necessary, so the part "here with a wavelength of 0.64 μm" will be omitted in the revised version.

12. L189: this begs the question as to why are some variables considered "products" while other variables are considered "extra data layers"? What is the difference? Why not report everything as "products"?

**Author reply:** The products as described in Table 2 are the main variables that are provided, each in separate files. In most cases the files contain additional variables, which we call "data layers" here, that are related to the main variables. For example, in the CFC file, also low-, medium- and high-level cloud fractions are provided. Similarly, in the LWP file, COT of liquid clouds is provided. Since the term "data layer" may be confusing, we will skip it, so the sentence becomes: "Both COT and CRE are available in the product files for LWP and IWP."

13. L191: by implication, a CGT implies a cloud base height (CBH). When researchers use this product, they may very well use it to infer a CBH. Have you examined CGH/CBH?

**Author reply:** This is in principle true. In addition, with the respective error estimates of CTH and CGT, one could also derive an error estimate of CBH. However, we have not done any validation of CBH derived in this way. It would be an interesting topic, which we may analyze in the future or which a user of the data could investigate.

14. L194: probably should clarify with '…separately for both liquid and ice clouds'. Is this correct?

**Author reply:** Yes, the word "separately" will be added in the revised version.

15. Same paragraph: May want to cite documentation or papers for MODIS L3 gridded products that provide similar histograms

    **Author reply:** The sentence "Similar Joint Cloud Histograms are provided in the ISCCP and MODIS data sets (Platnick et al. 2015)." will be added at the end of this paragraph.

    Added reference:

    Platnick, S., King, M. D., Meyer, K. G., Wind, G., Amarasinghe, N., Marchant, B., Arnold, G. T., Zhang, Z., Hubanks, P. A., Ridgway, B., and Riédi, J.: MODIS Cloud Optical Properties: User Guide for the Collection 6 Level-2 MOD06/MYD06 Product and Associated Level-3 Datasets, Version 1.0, available at: https://modis-images.gsfc.nasa.gov/_docs/C6MOD06OPUserGuide.pdf (last access: 7 July 2023), 2015.

16. L220: what is the definition of 'moisture content'? Vapor only? In the column? Or does it include the sum of vapor, cloud water content, and precipitation?

    **Author reply:** We suggest the following reformulation:

    " …total atmospheric moisture content (i.e., column-integrated water vapour excluding cloud water and precipitation),…."

17. L232: 'instead' rather than 'in its stead' **Author reply:** To be changed.

18. L238: Why 231K (-42C) instead of 233K (-40C)? Why 265K (-7C) rather than some other threshold? Can you add a reference(s) to justify these choices?

    **Author reply:** Selection of these thresholds was motivated based on the temperature limits where liquid droplets and ice crystals occur in supercooled clouds, which are included in the extended cloud phase retrieval. These numbers do not match exactly respective limits given in the literature: -40C (e.g. Tabazadeh et al. 2003) and -6C (e.g. Hobbs and Rangno, 1985). For the CPP retrieval algorithm, the specific limits were selected empirically based on comparisons with observations from Cloudsat and CALIPSO.

    Added references:

Hobbs, P. V. and Rangno, A. L.: Ice Particle Concentrations in Clouds, J. Atmos. Sci., 42, 2523–2549, 1985.

Tabazadeh, A., Djikaev, Y. S., and Reiss, H.: Surface crystallization of supercooled water in clouds, P. Natl. Acad. Sci. USA, 99, 15 873–15 878, 2003.

19. L253: I did not see any presentation or discussion of uncertainty estimates in this paper. Are we talking about uncertainties in the ancillary data that are somehow utilized to produce the products in Table 2? Or are these uncertainties on the products themselves reported in Table 2? In either case, these should be described, cited, shown in a figure, their methodological approach described and cited, and summarized in some way that demonstrates their use.

**Author reply:** The approach for estimating uncertainties depends on the product, and including figures to show or validate the uncertainties would make the paper too long. However, we agree that more information on the derivation of the uncertainty estimates should be provided, especially since such estimates have been added to several of the products which lacked such information in the predecessor CLARA-A2. Thus, we can say the following:

- The CFC product is derived from cloud probability products (CMAPROB) where uncertainties are implicitly embedded in the product itself (i.e., maximum reliability at 0% cloud probability and at 100% cloud probability but with maximum uncertainty at 50% cloud probability). These cloud probabilities are available in the Level 2b products so that any user could define their own confidence levels depending on applications. For the L3 product, a simple measure is calculated which is based on the averaging of the 'distance' to the 50% cloud probability level for all pixels within the 0.25 degree grid box. We propose to add the following sentences after line 226:

  "CFC uncertainties for the Level 2b product can be interpreted directly from the CMAPROB product which is provided together with the binary cloud mask. Maximum uncertainty is found at the 50% cloud probability level. For the level-3 product, a simple estimation based on the averaging of the probability distance from the 50% threshold for clear and cloudy pixels is provided."

- The CTO product (Level 2) has an uncertainty estimate based on Quantile Regression Neural Networks. This has already been described

on lines 232-233. The estimated 16[th] and 84[th] percentiles correspond to one sigma level. For the L3 product, The CTO 1-sigma uncertainties from Level-2b files are propagated into Level-3 following Stengel et al. (2017) as summarized in the cloud products ATBD (available via the doi link in Sect. 5) in Section 4.4.8. We propose the following sentences after line 233:

"For estimating the CTO uncertainty in the level-2b product, the average of the absolute CTO difference from the 16[th] and the 84[th] percentile is provided. These CTO 1-sigma uncertainties from Level-2b files are then propagated into level-3 products, following Stengel et al. (2017)".

Reference added:

"Stengel, M., Stapelberg, S., Sus, O., Schlundt, C., Poulsen, C., Thomas, G., Christensen, M., Carbajal Henken, C., Preusker, R., Fischer, J., Devasthale, A., Willén, U., Karlsson, K.-G., McGarragh, G. R., Proud, S., Povey, A. C., Grainger, R. G., Meirink, J. F., Feofilov, A., Bennartz, R., Bojanowski, J. S., and Hollmann, R.: Cloud property datasets retrieved from AVHRR, MODIS, AATSR and MERIS in the framework of the Cloud_cci project, Earth Syst. Sci. Data, 9, 881-904, https://doi.org/10.5194/essd-9-881-2017, 2017."

- For the LWP, IWP, CDNC and CGT products, we will add a brief description in the relevant section (3.1.1), and the reader will be referred to the ATBDs for more details. Specifically, we will add the following at L250: "Estimated uncertainties in reflectance measurements and various input variables (e.g., surface albedo, total ozone column) are propagated to yield uncertainty estimates in retrieved COT and CRE. These are, in turn, propagated to uncertainty estimates in LWP, IWP, CDNC and CGT. Further details are given by Stengel et al. (2017) and in NWC SAF (2021)."

Added reference:

NWC SAF: Algorithm Theoretical Basis Document for Cloud Micro Physics of the NWC/PPS, Satellite Application Facility on Nowcasting and Very Short Range Forecasting, NWC/CDOP3/PPS/SMHI/SCI/ATBD/CMIC, Issue 3.0, 12 October 2021.

- For the TOA products (OLR and RSF), no uncertainties were a priori calculated. Instead, the uncertainty in a given grid box is estimated a posteriori, by validating the data against the state-of-the-art CERES data record (considered as the "golden standard" among the TOA radiative flux data records). It provides the user a good idea of the performance, expressed as Mean Absolute Bias on daily and monthly time scale. It may vary in time and space, and all this information is available to the user in the Validation Report. We can meet the reviewer's request by introducing a table containing the MAB w.r.t. CERES on both daily and monthly timescales, for both OLR and RSF.

- For the surface albedo product we can add the following paragraph after line 506:

  "The albedo data do not contain uncertainty estimates per grid cell. However, a wide variety of parameters describing the statistical distribution and sampling density of the retrieved albedos are provided in the data files, e.g. skewness, kurtosis, and number of valid observations per grid cell."

  For the remaining CLARA-A3 products, no specific uncertainty parameters are provided with the Level 3 product but any user has access to extensive information in the validation reports describing the overall product quality.

- For the surface radiation products there are no uncertainty estimations per grid cell. However, the extensive comparisons with measurements at BRSN stations (detailed in the referenced validation report) should give a good view of the uncertainties of this product. In our opinion, the description in the text is sufficient.

20. L264: do not need 'in addition' at the end of the sentence **Author reply:** Yes, will be removed.

21. L279: the phrasing is a bit awkward. Is this supposed to mean 'more thin cloud with a COT < 0.3 is detected' or 'more thin cloud with a COT > 0.3 is detected'?

    **Author reply:** We suggest the following revised formulation at line 278 that hopefully can be clearer:

    "…CALIPSO-Passive results. This indicates that a substantial fraction of all clouds with an optical thickness less than 0.3 are now detected in

CLARA-A3. This clearly differs from the performance of the predecessor CLARA-A2 where less thin clouds were detected."

22. L292: 'significant trends' implies that a formal significance test in the trends was performed. Do the authors mean that there are no trends with 95% significance? How did you do this test or conclude this result?

   **Author reply:** It is clear that one has to be careful when using the word "significant" in a manuscript, knowing about the strict statistical meaning of this word. We will change the word "significant" here to the more general and descriptive term "clear" in the text. However, true significance tests have actually been made and published recently (https://www.preprints.org/manuscript/202306.1668/v1). Such studies should preferably be based on de-seasonalized data which was also done in the referenced paper. Thus, we will emphasize in the text that results from more detailed trend studies can be found in the referenced paper.

23. Figure 4: Why does MODIS Aqua show a decrease in liquid cloud? Is this a calibration issue or a physical change in cloud properties?

   **Author reply:** This was not investigated further. The same decrease does not appear in CLARA-A3 and ESA-CCI.

24. L330: Why specifically should we care to read the report? It would help to explain what is in it at a very high level. It is not useful to tell the reader that they need to read another report or series of papers to get the gist of the current paper. The information should be self-contained in some manner.

   **Author reply:** We have reported our most important results from validation studies for all products but there are a lot more details to find in the referenced validation reports. To really go into details describing all validation results for all CLARA-A3 products in the current manuscript would totally explode the size of it. At the same time, we think it is fair to the reader to inform about the more extensive validation reports as well as already published separate studies of the CLARA-A3 products. So, we think we have a reasonable balance between the two objectives to describe the CDR content and to briefly describe product quality. After all, the goal of the ESSD journal is to present new science datasets, not to analyze each of them in depth. The latter should be done in separate publications.

In conclusion, the reader does not need to read the validation reports to get the gist of the paper. But the reports contain additional, more detailed information on the validation of CLARA-A3. We will rephrase as follows: "Additional details on validation results are provided in the dedicated CLARA-A3 Cloud Products validation report available through the link given in Sect. 5.".

25. Same part of manuscript: What about trends in IWP, LWP, and effective radius for liquid and ice clouds? Have these been examined?

    **Author reply:** They have been examined in terms of the stability of their bias with the reference data sets. Although time series are mostly stable over the full CLARA time range, there are signatures of satellite switches and orbital drift. Extensive results are given in the validation report (CM SAF, 2023).

26. L340: spell out Table **Author reply:** OK.

27. L361: look-up table should be spelled out on first use. Also, need to add space between 'LUT' and 'that' **Author reply:** Yes, we will change this.

28. L363: add Hersbach reference for ERA5. Also, define ERA5T. Is this surface temperature? Or something else?

    **Author reply:** OK, we will add this reference also here (even if it was introduced already on line 221). Then, we realize that we started using the term ERA5T before it was properly explained (which is done on lines 660-662). The easiest way to solve it is to just state "(i.e., a preliminary version of ERA5, see Sect. 3.5 for further explanation)".

29. Paragraph starting on line 380: It is unclear what we should take home from the use of a monthly climatology for aerosol averaged over many years. How does this impact the interpretation of the solar products in CLARA-A3 on a daily, instantaneous basis? Relative to the total magnitude of the value of the products, how large is the contribution from aerosol? Less than a few %? Greater than that by a little, or a lot?

    **Author reply:** The exact contribution of the aerosol is difficult to determine (and not part of the current study). The uncertainty of the daily (and instantaneous, even though not provided) surface solar radiation data is impacted by many contributions, incl. the use of monthly aerosol information, but also e.g. the limited number of available satellite observations to estimate the daily average. Since the main determining

factor of daily surface solar radiation is cloudiness, we expect the impact of using a monthly aerosol climatology on the daily surface irradiance to be less than a few %. There surely, however, are situations, when the impact is larger, e.g., very high dust loadings under clear sky conditions. This limitation on the use of monthly aerosol information for the analysis of daily surface radiation data will be mentioned in the revised version of the manuscript.

30. L391: end of sentence, also should be noted that it does not contain day-to-day variability, which conceivably can be significant

   **Author reply:** We agree with the reviewer that (possibly significant) day-to-day aerosol variability is not included in the clear-sky surface irradiance; this will be stated in the revised version of the manuscript.

31. L406: 'The grid boxes…' **Author reply:** Yes, we will change.

32. L407: end of sentence repetitive **Author reply:** This comment is not clear, we do not see any repetitive statement here. However, we will move the remark on the difficulty for using the CLARA-A3 data record to estimate global averages (line 404 to line 407) at the end of this paragraph, since it also applies to the surface net solar radiation.

33. L419 and L422: Is ERA5T the same as ERA5(T)? **Author reply:** Yes. We will change so we use ERA5T everywhere.

34. L530: 'was very good' is not entirely useful or quantitative

   **Author reply:** We propose removing the qualitative part "we note that the CDR performance was very good regarding mean bias and stability (the temporal trend in bias)." and reformulate in order that the quantitative part is highlighted first: "As a summary of the results, the mean relative bias was…".

35. L562: how many ADMs are used?

   **Author reply:** In total, there are 600 scene types, so that is the amount of ADM's that were potentially used. Evidently, their relative importance depends on the occurrence of the scene types.

36. Figure 10: The fact that ERA5 and CLARA-A3 track so closely at short and long temporal scales could be interpreted in a way that suggests CLARA-A3 depends heavily on ERA5 ancillary inputs, and that you are

basically making another version of ERA5. Is this fair to say or is there a clear advantage to using AVHRR data for the solar and thermal flux and irradiance products? This gets back to the earlier comment about why are these products being generated in the first place? What is the added value over other climate records?

**Author reply:** What you see in Figure 10 is only the global mean RSF, which gives an indication of the record's stability as mentioned in L615. However, that's only one aspect of a data record's performance: also important is the regional uncertainty, which gives an indication of the bias on a particular time in a particular grid box: this is much worse for ERA5 reanalysis since it is basically a climate model constrained with observations (but not with TOA radiative flux observations). But more in general, about the added value of the climate records: we agree with the reviewer that this issue deserves some more attention. We therefore add the following paragraph which addresses the remark of the reviewer:

"A full global coverage of broadband observations is provided by the Clouds and the Earth's Radiant Energy System (CERES) instruments and derived products (Loeb et al., 2018), which are acknowledged to be the golden standard w.r.t. radiative flux data records. However, there has been an increasing need for long-term, high-resolution TOA albedo products in monitoring the climate impacts of regional-scale events such as air pollution, urbanization, forest fires, and other small-scale land cover changes (Song et al., 2018), which can hardly be detected from data sets with coarse spatial resolution (Wang et al., 2016), and small-scale atmospheric processes e.g. valley fog (Clerbaux et al., 2009). Furthermore, in absence of a global long-term CERES-like CDR, many studies focusing on long term model validation or trend detection fall back to "surrogate datasets" such as reanalysis (e.g. ERA-Interim) or radiative transfer computations (e.g. ISSCP), but would otherwise have preferred a more observation-based alternative. Concerning CERES, two limitations can thus be identified: (1) the products are relatively recent, e.g. starting in year 2000 for the EBAF product, and (2) the products have a relatively coarse spatial resolution of 1°x1° (Fig. 9a). The currently developed TOA flux products in CLARA-A3 resolve those two drawbacks, respectively by (1) a prolongation back in time to the late 70's and (2) by increasing the spatial resolution to 0.25°x0.25° (Fig. 9b). A third advantage of the new CDR's lies in their synergy and compatibility with the other CDR's from the CM SAF CLARA product family (cloud mask and other cloud parameters, surface radiation, surface albedo, etc.) sharing common algorithms and processing chains."

The mentioned references above will be added to the manuscript.

37. L619: did not see Canty et al in reference list

   **Author reply:** Indeed, it is missing in the reference list. The following reference will be added to the list:

   Canty, T., Mascioli, N. R., Smarte, M. D., & Salawitch, R. J. (2013). An empirical model of global climate–Part 1: A critical evaluation of volcanic cooling. Atmospheric Chemistry and Physics, 13(8), 3997-4031.

38. Figure 11: Why show MAB instead of actual values of OLR? It would be useful to consider showing OLR in the same way as you did with fluxes of RSF in Figure 10.

   **Author reply:** The conclusions of global mean OLR validation are shortly described in the text (L625-627), just like also has been done for the RSF MAB (L635-637). Figures 10 and 11 should be understood as non-exhaustive (illustrative) examples of the TOA flux validation. As mentioned in L607-608, it is not the intention to include the complete validation for the TOA fluxes in this article because it would go beyond its scope and there are of course limitations on article length and figure count (which is the same approach as for the other products). We simply wanted to show what kind of validation methods were used, i.e. bias and temporal evolution of bias (for stability) and Mean Absolute Bias (for regional uncertainty). There are a number of other validation aspects that we could show as well (e.g. daily mean analysis, monthly mean diurnal cycle, etc..) which are, however, not shown here for the same reasons. So instead, we refer to the TOA flux validation report (119 pages) which is publicly available on the DOI landing page of the product and contains the full validation on all aspects and with all details.

39. L707: 'It is worth…' **Author reply:** Yes, we will change.

40. L711: rogue letter 't' **Author reply:** Yes, we will remove it.

41. L733: If the effective radius is not shown in the paper, at least point to this result in the validation report or elsewhere. This is interesting but the microphysics haven't been shown in the paper.

   **Author reply:** As mentioned before, the validation report (CM SAF, 2023) contains extensive comparisons of time series of cloud products with other datasets, including CLARA-A2. Specifically, Figure 5-36 in

the validation report shows that the cloud droplet effective radius time series is much more stable than in CLARA-A2. We have added "(see Figure 5-36 in the CLARA-A3 Cloud Products validation report, available through the link given in Sect. 5)." at the end of the paragraph.

---

## Author Comment (AC3)

Reply to Referee #3 on the Earth System Science Datasets manuscript

**" CLARA-A3: The third edition of the AVHRR-based CM SAF climate data record on clouds, radiation and surface albedo covering the period 1979 to 2023"**

**by**
**Karlsson et al, 2023**

**Repeating general comment:**

**The authors present some results from the CLARA-A3 validation against other similar cloud and radiation products. Production of this dataset is a significant accomplishment and an important addition to global climate studies of cloud and radiation. It is also clear from the link in Section 5 that there is a great deal of documentation and validation supporting this paper, which therefore necessitates selection of a few 'highlights' to be included here. That said I believe there are areas that require more detail in order for this to be considered useful as a stand-alone article.**

**Reply:**

We thank the referee for these encouraging words. Answers to the referee's comments and questions are given below together with suggestions on how to improve and update the manuscript.

**Detailed comments:**

1. L180: My understanding is there is no attempt at multilayer detection, and the cloud top phase is assumed to extend throughout the cloud column when deciding whether to calculate IWP or LWP. In that case I assume there is additional uncertainty in IWP estimates where it is unknown when ice overlaps water? It would be nice to know how uncertainty is estimated for these products and whether multi-layer clouds translate to higher uncertainty.

   **Author reply:** There is indeed no multilayer detection and for LWP/IWP calculation a single phase throughout the column is assumed. The presence of multiple phases, typically when low liquid clouds reside below high ice clouds, leads to an error in the optical and microphysical cloud property retrievals. However, the uncertainty estimates do not

include deviations from the assumption of horizontal and vertical homogeneity of the clouds as a source of error, and therefore such deviations will not be reflected in the uncertainty estimates.

We will try to update the text for clarifying these issues.

Changes in revised manuscript with tracked changes:
- Lines 188-189
- Lines 277-279

2. L252: The description of CPH, COT, and CRE are very brief, and we don't see any figures showing results from these products. Have the updates listed here resulted in lower uncertainty, or simply higher confidence in the reported uncertainty? Some numbers describing the changes in uncertainty for these products would be helpful.

**Author reply:** The descriptions of the CPH, COT and CRE retrieval are of similar length as the other cloud products (CMAPROB, CTO). Because of the many products included in CLARA-A3, these descriptions need to be brief in order not to further blow up the size of an already extensive manuscript. The same holds for the number of figures. However, there are figures showing CPH (Fig. 4) and LWP/IWP (Fig. 5), which is in fact a combination of COT and CRE. For CPH, clear improvements of the product are visualized in Fig. 4, in the sense that CLARA-3 results agree better than CLARA-A2 with state-of-the-art results from MODIS (if ignoring the negative trend in MODIS results in recent years which we for the moment do not know the explanation). For LWP/IWP results in Fig. 5, results are also a bit closer to MODIS results, at least for the IWP product. The update in L253 has presumably made the uncertainty estimates more realistic because more error sources, such as surface reflectance and atmospheric water vapor column, have been taken into account, although this is very hard to prove. We will emphasize this better in the text. The overall magnitude of uncertainty estimates has not changed much compared with CLARA-A2.

Changes in revised manuscript with tracked changes:
- Lines 283-285

3. Figure 2: There is a great deal of overlap making it difficult to differentiate among the records. Perhaps an anomaly plot would help? If it were deseasonalized we could also determine whether the records showed different trends over time.

**Author reply:** Yes, we are aware of this problem but still think it could be useful to give an overall plot of all original data despite these weaknesses. Instead of changing this plot, we suggest to refer to another paper where both anomaly plots and deseasonalized plots are presented and discussed more in depth. That paper had a quicker review process than this manuscript which means that it is already published.

Deseasonalized anomaly plots are available here (Figure 6 in the mentioned paper):

Devasthale, A.; Karlsson, K.-G. Decadal Stability and Trends in the Global Cloud Amount and Cloud Top Temperature in the Satellite-Based Climate Data Records. Remote Sens. 2023, 15, 3819. https://doi.org/10.3390/rs15153819
A corresponding plot is also presented for global mean cloud top temperature in the same paper (Figure 7).

Changes in revised manuscript with tracked changes:
- Line 330
- Lines 337-338.

4. L292: Was a statistical test applied to the CTP records in Figure 3 to deduce there are no significant trends? If so could that information be included here?

   **Author reply:** This has been done. Results were published in the paper referred to in the previous point but only for the cloud top temperature parameter, not CTP. However, CTT and CTP are strictly related (as is mentioned on line 166 in the manuscript). We will refer to that paper on this topic.

   Changes in revised manuscript with tracked changes:
   - Line 330
   - Lines 337-338

5. Figure 4: Is the liquid cloud fraction a percentage of total cloudiness phase (i.e. liquid / liquid + ice)?

   **Author reply:** The liquid cloud fraction in Fig. 4 is defined here as the liquid cloud amount relative to the total cloud amount, i.e. liquid / (liquid + ice). This information will be added to the caption in the revised manuscript.

Changes in revised manuscript with tracked changes:
- Line 355

6. L380-391: I find this paragraph confusing. When are the instantaneous versus daily irradiance calculations used? Is one for Level 2 and one for Level 3? Also, an aerosol record that extends from 1979-2025 is used, but only to create a 12-month climatology? Is this also the case for atmospheric gases? For example, is $CO_2$ rise accounted for?

**Author reply:** We are sorry for the confusion and try to explain better here.

The instantaneous (all sky and clear sky) and the daily (clear sky) irradiance values are used to derive the daily all sky irradiance data (i.e., the final product of the data record) via formula (1). This formula allows the accurate estimation of all-sky daily irradiance by weighting the clear-sky daily irradiance (derived from a RTM) with the ratio of the sum of the instantaneous all-sky (derived from the satellite retrieval) and the sum of the instantaneous clear-sky (derived from a RTM) irradiance. Different aerosol information is used for the instantaneous and the daily irradiance estimations. As formula (1) includes the ratio of the 2 instantaneous irradiance values (all-sky and clear-sky) the aerosol impact on the instantaneous irradiance on the daily mean surface irradiance is limited; the aerosol effects in the daily all sky irradiance data are dominated by those represented in the estimation of the clear sky daily irradiance.

The clear-sky daily irradiance is used for the generation of the 'Level 3' daily averaged surface irradiance. Thus, the instantaneous surface irradiances (clear-sky and all-sky) could be considered 'Level 2'-products, even though the terminology in this case is not well defined (and no Level 2-type products are accessible for users).

The aerosol data from 1979 to 2020 was used to estimate a monthly climatology, which was used for the estimation of the clear-sky daily mean irradiance. We consider the temporal variability and the long-term trend of the aerosol optical depth, which was derived from model simulations, to be only of moderate accuracy; to limit the impact of the aerosol information (with unknown / moderate accuracy) on the final surface irradiance data record we decided to only use the climatological information in the surface irradiance estimation; we acknowledge that the direct aerosol effect is not considered in the final surface irradiance data

record; this fact should be considered in the interpretation of longer-term variability and trends derived from the CLARA-A3 SIS data record.

For atmospheric gases, in particular water vapor and ozone, instantaneous and daily data are used for the estimation of the instantaneous and the daily surface irradiance, respectively. The impact of $CO_2$ on the surface solar irradiance is negligible and is not considered; for the longwave surface radiation, the increase in $CO_2$ accounted for based on ERA-5.

We think that a lot of this information is, in fact, already described in the text. In the interest of keeping the size of the text not too long, we did not modify the text further but hope that it is enough with this explanation.

7. L592: How is the instantaneous OLR estimated for the AVHRR/1 where channels 4 and 5 are the same?

**Author reply:** The instantaneous OLR is estimated from AVHRR/2 and AVHRR/3 using regression fits with 6 predictors (i.e. 7 coefficients), among which are the channel 4 and channel 5 brightness temperatures. For AVHRR/1, there is a different set of regression fits which do not make use of the channel 5 brightness temperature, resulting in only 4 predictors (5 coefficients). The article Clerbaux et al. (2020), cited at the end of that paragraph on line 599, describes this in full (including a validation of its impact). We do agree with the reviewer that it is not clear for the reader, and will therefore add the following clarification on line 593: "This is done by regressions on the same large database of collocated AVHRR-CERES observations (as used for the RSF); for AVHRR/1 the regressions only make use of the channel 4 brightness temperature, for AVHRR/2 and AVHRR/3 both channel 4 and 5 are used (Clerbaux et al., 2020)."

Changes in revised manuscript with tracked changes:
- Lines 666-669

**Editorial comments:**

L114: 'Full' shouldn't be capitalized. **Author reply:** Yes, will be corrected.

Changes in revised manuscript with tracked changes:
- Line 117

L264: 'in addition' is repetitive. **Author reply:** Yes, will be corrected.

Changes in revised manuscript with tracked changes:
- Lines 295-296

L361: There should be a space between 'LUT' and 'that' **Author reply:** Yes, will be corrected.

Changes in revised manuscript with tracked changes:
- Line 409

L566: 'then' should be capitalized. **Author reply:** Yes, will be corrected.

Changes in revised manuscript with tracked changes:
- Line 640

L712: There is an extraneous 't' after CLARA-A3. **Author reply:** Yes, will be corrected.

Changes in revised manuscript with tracked changes:
- Line 798

---

## Author Response (AR2)

Dear Editors,

We are pleased with this decision and we will now upload the files needed for the publication process. However, I think I need to comment the requirement to check further figures for colour blindness/deficiencies. We have not ignored this requirement, not at all. This has been a discussion point all through the manuscript writing and several figures have been revised for that purpose before submitting the manuscript.

I also used the Coblis tool directly after the files were validated and I could not at that point find anything serious. I concluded then that all figures basically could be interpreted reasonably well even with the simulated colour deficiencies. So, I took no further action and waited for the final editor decision.

But, I admit that it is sometimes hard to judge the effects of various colour deficiencies. Further, there is some kind of trade-off limit if you restrict colours used too much. I mean, to skip completely using the blue colour to help those with Tritanopia (which is extremely rare) will seriously limit what you can do if you want to use a colour representation. This is a problem when a monochromatic view is clearly not able to show the things you want.

Since the red-green colour deficiency is the most common deficiency (in fact, I have myself a ~25% red-green deficiency!), we have basically tried to avoid colour combinations where those two colours are present simultaneously in the same plot (at least, if they show things that are really important for the discussion in the text). When checking for the red-green deficiencies using Coblis I could not see any big remaining problems due to this in the plots.

I think I need to go through all 11 figures with specific comments to explain our thoughts and choices of colours:

Figure 1:

This is an important figure for the article showing the use of data from 16 satellites during 42 years of measurements. Each satellite curve has been given a specific colour. In this case there are also co-existing red and green colours since it is a bit difficult to find 16 different colours and still avoiding red and green. However, we think we can be excused in this case since we also have written clearly in the figure the name of the satellite in front of each curve. With this backup-text we don't think this can be misinterpreted.

Figure 2:

Here we again have a large number of curves (8 curves) with different colours. In this case we do emphasize two curves with thick lines, i.e. the ones for CLARA-A2 (blue) and CLARA-A3(red). We want to show the change when going from CLARA-A2 to CLARA-A3. This is the main message. But we also wanted to set the CLARA-results in perspective to other available datasets without really allowing full

details (i.e., these curves are partly obscured by the CLARA curves). Here we can see that the CLARA-A3 results are now in better agreement with the bulk of other datasets whereas CLARA-A2 was clearly on the low side. Notice that we avoided simultaneous use of red-green for the two thick lines.

Figure 3:

This is basically a similar plot as in Figure 3 for another parameter but since the curves are better separated here we skipped the use of thicker lines for CLARA-A2 and CLARA-A3. I could not find any particular problem with this visualization using Coblis. If a reader also use the magnifying tool (zoom) I cannot see that the figure can be misinterpreted.

Figure 4:

Basically the same conclusions as for Figure 3. Coblis did not indicate a problem when testing the red-green deficiencies. For us, the separation of CLARA-A2 and CLARA-A3 is most important and this seems to work reasonably well.

Figure 5:

Now we are presenting spatial (global) plots. Here we have used two-colour schemes (i.e., going from dark blue/green to yellow or from blue to red). This is at least much better than if using more complicated colour schemes with more colours (like rainbow or jet). And I think the combination of blue and red works nicely to show the most important features. Coblis did not indicate any big problems here.

Figure 6:

Again, a spatial plot using a two-colour scheme going from dark blue to yellow. Coblis did not indicate any problems here.

Figure 7:

Basically the same comment as for Figure 6. However, here we use two different two-colour schemes: one going from red to black (via white in the middle) for circle symbols and one going from dark blue to yellow for the global plot. I think that it worked well with different Coblis options but here I am a bit more uncertain. However, after discussing it with the other authors we still think this is the most useful option. The changed validation results ( for rMBE) are also thoroughly commented in the text and the reader should not misinterpret this message, we think.

Figure 8:

This simple schematic plot cannot possibly be misinterpreted, even with colour deficiencies. Tests with Coblis did not show any problems.

Figure 9:

These regional plots caused us quite a lot of problems initially when other colour schemes were used. But this colour scheme seems to work well according to Coblis. Thus, I think we have found a useful representation in the end.

Figures 10 and 11:

Again we have line plots with several (i.e., five or six different) datasets. We think that colour schemes are not the biggest problem here (at least according to Coblis) but rather the visualization of all the details. But if using the magnifying tool (zoom) the reader would certainly be able to identify and separate these datasets. We also found it important to really show results over the full 42-year period which consequently creates a lot of details.

With this I hope I have shown that we have taken the colour blindness/deficiency problem seriously and done the best we can to minimize them. If you disagree, we would appreciate to get some more specific instructions of what can be done to improve things further.

Best regards

Karl-Göran Karlsson (on behalf of authors)

PS. The reply to Reviewer 3 is now uploaded to the Interactive Discussion part.